# Local Hessian Spectral Filtering for Robust Intrinsic Dimension Estimation

**Genki Osada** [1]

## Abstract

While diffusion models enable new approaches for estimating Local Intrinsic Dimension (LID), existing methods fail in high-dimensional spaces where noise from vast normal directions overwhelms the tangent signal. We propose Local Hessian Spectral Dimension (LHSD), which resolves this by applying spectral filtering to the log-density Hessian, explicitly cutting off large eigenvalues associated with normal directions to count zero-curvature tangent directions. Implemented using Stochastic Lanczos Quadrature (SLQ), LHSD avoids full Hessian construction, achieving linear scalability with dimension $D$. Experiments on synthetic and real data confirm LHSD's superior robustness and its utility in detecting memorization in large-scale diffusion models.

## 1. Introduction

The manifold hypothesis posits that high-dimensional data concentrate along lower-dimensional structure, *manifolds*, providing a foundation for deep learning (Belkin et al., 2006). To quantify the geometric complexity of these structures, the Local Intrinsic Dimension (LID) estimates the local degrees of freedom (Ozakin & Gray, 2009; Brown et al., 2023). This metric is vital for analyzing generalization (Ansuini et al., 2019), detecting anomalies (Kamkari et al., 2024a) and adversarial examples (Ma et al., 2018), and recently, explaining *memorization* in generative models (Ross et al., 2025), where a model inadvertently generates exact copies of training samples. However, accurately and efficiently estimating LID for high-dimensional data (e.g., images with $D > 1000$) remains a challenge.

Historically, $k$-nearest neighbor (kNN) based approaches were the standard (Cayton, 2005). While effective in low-dimensional settings, prior kNN-based LID estimators have

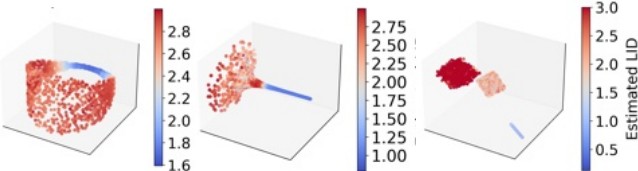

*Figure 1.* Synthetic Manifold: Moon (left), Funnel (center), and $\mathcal{L}^{1+2+3}$ (right). The $\mathcal{L}^{1+2+3}$ dataset consists of 3D cube, 2D square, and 1D line submanifolds. The overlaid colors indicate the LID estimated by our LHSD, accurately recovering the underlying manifold dimensionalities.

been observed to deteriorate empirically as the ambient dimension increases (Tempczyk et al., 2022; Kamkari et al., 2024b; Tempczyk et al., 2026). Classical analyses also suggest that reliable estimation in high dimensions may require very large sample sizes (Levina & Bickel, 2004). This makes accurate and efficient LID estimation for high-dimensional data challenging in practice.

To address this, approaches utilizing deep generative models, particularly diffusion models, have emerged (Horvat & Pfister, 2021; 2022; Tempczyk et al., 2025). By leveraging geometric information derived from the learned score function (the gradient of the log-density), these methods avoid neighborhood search, offering a promising path toward scalable estimation in high-dimensional data.

However, Tempczyk et al. (2026) highlighted that these methods fail in high dimensions, particularly in *high co-dimension* settings typical of image data, where the ambient dimension far exceeds the manifold dimension. Since LID is equivalent to the dimensionality of the tangent space, distinguishing the tangent component (signal) from the normal component (noise) is crucial for accurate estimation. However, existing methods indiscriminately sum values, such as score norms, across all dimensions. This severely degrades the S/N ratio: geometrically, the sharp density drop in the normal direction yields magnitudes that grow unboundedly, overwhelming the tangent signal, especially in high co-dimension settings where the normal subspace is vast. This failure to separate noise from signal leads to significant estimation errors, as quantified in Sec. 4.

In this work, we propose Local Hessian Spectral Dimension (LHSD) to address this issue. LHSD identifies LID by counting zero-curvature components in the log-density

[1]LY Corporation. Correspondence to: Genki Osada <genki.osada@lycorp.co.jp>.

*Proceedings of the 43rd International Conference on Machine Learning*, Seoul, South Korea. PMLR 306, 2026. Copyright 2026 by the author(s).

Hessian, corresponding to tangent directions. To do this robustly, we introduce *spectral filtering* using a smooth filter to map eigenvalues to $[0, 1]$. This explicitly cuts off massive values from normal directions while preserving tangent counts. A major challenge here is the prohibitive cost of constructing the full Hessian matrix. We address this challenge via Stochastic Lanczos Quadrature (SLQ), enabling estimation with computational cost linear in dimension $D$. Our main contributions are:

- **Robust LID Estimator:** LHSD selectively counts tangent dimensions via Hessian spectral filtering. By suppressing normal eigenvalues, it ensures robustness in high dimensions.

- **Verifiable Hyperparameter Selection:** Overlaying the filter profile on eigenvalue distributions visually diagnoses parameter correctness and guides necessary adjustments.

- **Scalable Algorithm:** Integrating spectral filtering with SLQ bypasses explicit Hessian construction, achieving fast, linear complexity $\mathcal{O}(D)$.

- **Extensive Empirical Validation:** Experiments on synthetic and real data confirm LHSD's performance and utility in detecting diffusion model memorization.

## 2. Background and Related Works

### 2.1. Local Intrinsic Dimension (LID)

In machine learning, the manifold hypothesis is widely accepted (Belkin et al., 2006; Bengio et al., 2013). It posits that high-dimensional real-world data lie near a low-dimensional manifold ($d \ll D$) embedded within the ambient space $\mathbb{R}^D$. Local intrinsic dimension (LID) measures the manifold's local degrees of freedom, geometrically interpreted as the tangent space dimension at $\mathbf{x}$ (Ozakin & Gray, 2009; Brown et al., 2023). Beyond manifold and representation learning (Ansuini et al., 2019), LID estimation aids practical tasks like out-of-distribution (Kamkari et al., 2024a) and adversarial detection (Ma et al., 2018). Of particular recent interest is the analysis of *memorization* in deep generative models (Carlini et al., 2023; Jeon et al., 2025). Prior work reports that verbatim training copies exhibit anomalously low LID, establishing it as a promising indicator for detecting memorization (Ross et al., 2025).

**kNN-based LID Estimation.** Since the shape and dimension of a manifold cannot be determined from a single data sample, LID estimation generally requires geometric information from the neighborhood of $\mathbf{x}$. Traditional estimators (e.g., MLE (Levina & Bickel, 2004), TwoNN (Facco et al., 2017), LPCA (Fukunaga & Olsen, 1971), ESS (Johnsson et al., 2015)) estimate the dimension based on sample proximity using $k$-nearest neighbors (kNN). While these methods yield asymptotically unbiased estimators for low-dimensional data, they face the curse of dimensionality in high-dimensional settings. As the volume of the space increases explosively, the concept of neighborhood becomes diluted, leading to a significant degradation in estimation accuracy with finite samples. We refer to these approaches collectively as kNN-based methods.

### 2.2. Diffusion-based LID Estimation

With the advancement of deep generative models, new methods have emerged that extract geometric information of the data manifold directly from learned score functions, without relying on neighborhood search as in kNN-based methods. In this section, we outline score-based diffusion models and existing LID estimators derived from them.

#### 2.2.1. SCORE-BASED DIFFUSION MODELS

Diffusion models generate data via a reverse denoising process. We consider data $\mathbf{x}_0 \sim p_0$ perturbed by Gaussian noise with scale $\sigma(t)$, resulting in the perturbed marginal distribution:

$$p_t(\mathbf{x}) = \int p_0(\mathbf{x}_0) \mathcal{N}(\mathbf{x}; \mathbf{x}_0, \sigma(t)^2 \mathbb{I}_D) \, d\mathbf{x}_0. \qquad (1)$$

A score model $\mathbf{s}_\theta(\mathbf{x}, t)$ is trained to approximate the score of this distribution, $\nabla_\mathbf{x} \log p_t(\mathbf{x})$, typically via denoising score matching (Hyvärinen, 2005; Vincent, 2011). (See App. A for details.) Our proposed LHSD derives the Hessian via $H(\mathbf{x}) = -\nabla_\mathbf{x} \mathbf{s}_\theta(\mathbf{x}, t)$ to analyze the spectral structure of Eq. (1). Existing methods use $\mathbf{s}_\theta$ in different ways, as described below.

#### 2.2.2. EXISTING METHODS

**LIDL** (Tempczyk et al., 2022). This method focuses on the relationship between Gaussian convolution and LID. It estimates LID by observing the decrease in density $p_t(\mathbf{x})$ relative to the increase in noise scale $\sigma(t)$ when $\sigma(t)$ is sufficiently small. Specifically, it is formulated as:

$$\text{LID}(\mathbf{x}) \approx D + \frac{\partial \log p_t(\mathbf{x})}{\partial \log \sigma(t)}. \qquad (2)$$

Although computational cost was a challenge in initial implementations using normalizing flows, efficient implementations using a single diffusion model have also been proposed (Kamkari et al., 2024b).

**FLIPD** (Kamkari et al., 2024b; Leung et al., 2025). Extending the concept of LIDL, FLIPD derives an analytical solution for $\partial \log p_t / \partial \log \sigma$ based on the Fokker-Planck equation, estimating LID in the following closed form:

$$\text{LID}(\mathbf{x}) \approx D + \sigma(t)^2 \left( \nabla \cdot \mathbf{s}_\theta(\mathbf{x}, t) + \|\mathbf{s}_\theta(\mathbf{x}, t)\|^2 \right). \qquad (3)$$

However, the term $\nabla \cdot \mathbf{s}_\theta$ indiscriminately sums *all* eigenvalues, inherently mixing tangent and normal components. This leads to estimation failure in high dimensions, where the massive normal curvature overwhelms the tangent signal (see Sec. 4). In contrast to such magnitude-based summation, our LHSD selectively *counts* only the tangent directions, ensuring robustness against normal noise.

**Normal Bundle (NB)** (Stanczuk et al., 2024). NB estimates LID via the dimension of the normal space. Since score vectors align with normal directions near the manifold, it constructs a matrix $S(\mathbf{x}) \in \mathbb{R}^{D \times M}$ by stacking $\mathbf{s}_\theta(\mathbf{x}, t)$ from $M$ noisy samples to compute:

$$\text{LID}(\mathbf{x}) \approx D - \text{rank}(S(\mathbf{x})). \quad (4)$$

However, rank estimation requires SVD. The resulting cost $\mathcal{O}(DM^2)$ scales to cubic complexity $\mathcal{O}(D^3)$ (given $M \propto D$), rendering NB intractable for high-dimensional data.

**Positioning of our LHSD.** While LIDL and FLIPD analyze density variations without separating tangent and normal components, NB and LHSD leverage geometric structure to explicitly identify the tangent dimensionality. Unlike NB's heavy SVD on first-order scores, LHSD leverages second-order Hessian spectra, using SLQ to bypass high costs and count tangent directions efficiently.

### 2.3. Hessian Spectral Analysis in Deep Learning

The validity of the proposed LHSD approach is supported by insights from Hessian analysis in deep learning (Ghorbani et al., 2019; Dauphin et al., 2014). It is known that the eigenvalue distribution of the Hessian in large neural networks often exhibits a characteristic structure consisting of a large *bulk* of eigenvalues concentrated near zero and a few large *outliers* (Sagun et al., 2017; Papyan, 2019). Genovese et al. (2014) and Ventura et al. (2025) established that the eigendecompositions of the density Hessian and the log-density Hessian, respectively, provide a natural decomposition of the ambient space into tangent and normal components with respect to an underlying manifold structure. Motivated by this geometric perspective, LHSD identifies and isolates the tangent components from the bulk using a Hill-type filter. This viewpoint is also related to Hessian LLE (HLLE) (Donoho & Grimes, 2003), which uses local Hessian structure for manifold learning and requires the manifold dimension $d$ as an input. In contrast, LHSD aims to identify tangent components directly from the score-Hessian spectrum.

## 3. Our Method

We propose Local Hessian Spectral Dimension (LHSD), which estimates LID by identifying zero-curvature tangent directions in the log-density Hessian. Leveraging the re-

---

**Algorithm 1** Local Hessian Spectral Dimension (LHSD)

**Input:** score model $\mathbf{s}_\theta(\mathbf{x}, t)$, noise variance $\sigma(t)^2$, probes $K$, cutoff $\kappa(t) \leftarrow c/\sigma(t)^2$, steepness $p$, Lanczos steps $m$
Define HVP oracle: $H(\mathbf{v}) := -\nabla_\mathbf{x}(\mathbf{s}_\theta(\mathbf{x}, t)^\top \mathbf{v})$
Initialize $\hat{d} \leftarrow 0$
**for** $k = 1$ to $K$ **do**
    Sample $\mathbf{v}_k \sim \{-1, 1\}^D$         ▷ Rademacher
    *// Stochastic Lanczos Quadrature (SLQ)*
    $T_k \leftarrow \text{Lanczos}(H, \mathbf{v}_k, m)$   ▷ Tridiagonal matrix
    $\{\tilde{\lambda}_j, \tau_j\}_{j=1}^m \leftarrow \text{Eigen}(T_k)$     ▷ Eigenpairs
    $\gamma_k \leftarrow \|\mathbf{v}_k\|_2^2 \sum_{j=1}^m \tau_j^2 f(\tilde{\lambda}_j; \sigma(t))$
    $\hat{d} \leftarrow \hat{d} + \gamma_k/K$
**end for**
**Return** $\hat{d}$              ▷ Estimated LID

---

sulting tangent–normal spectral separation, we formulate a soft estimator efficiently computed via Stochastic Lanczos Quadrature (SLQ).

**Tangent-Normal Decomposition.** Under the manifold hypothesis, data $\mathbf{x}_0$ concentrate near a smooth $d$-dimensional manifold $\mathcal{M} \subset \mathbb{R}^D$. When the data distribution $p_0$ is supported on $\mathcal{M}$, the perturbed distribution $p_t(\mathbf{x})$ (Eq. 1) can be approximated by integrating over the local neighborhood of the manifold. For a point $\mathbf{x}$ near $\mathcal{M}$ and sufficiently small $\sigma(t)$, the Gaussian kernel in the integral is maximized at the projection of $\mathbf{x}$ onto $\mathcal{M}$, denoted as $\mathbf{x}_\parallel$. Consequently, in local tangent–normal coordinates defined by the decomposition $\mathbf{x} = \mathbf{x}_\parallel + \mathbf{x}_\perp$, the log-density admits the asymptotic expansion:

$$\log p_t(\mathbf{x}) \approx \log p_\mathcal{M}(\mathbf{x}_\parallel) - \|\mathbf{x}_\perp\|^2/2\sigma(t)^2 + O(1), \quad (5)$$

where $p_\mathcal{M}$ is the density on the manifold, the term $-\|\mathbf{x}_\perp\|^2/2\sigma(t)^2$ is the log-density of a Gaussian with variance $\sigma(t)^2$ in the normal directions, and the $O(1)$ term accounts for normalization and higher-order curvature effects (Tempczyk et al., 2022). By taking the Hessian of the negative log-density, the quadratic term in the normal direction dominates the derivatives:

$$H(\mathbf{x}) := -\nabla^2 \log p_t(\mathbf{x}) = \Pi_{\text{nor}}(\mathbf{x})/\sigma(t)^2 + O(1). \quad (6)$$

Here, $\Pi_{\text{nor}}$ denotes the orthogonal projection onto the normal space of $\mathcal{M}$. This equation reveals that the curvature scales as $O(1/\sigma(t)^2)$ along normal directions, whereas tangent contributions from $\log p_\mathcal{M}$ remain $O(1)$ and become negligible in the small-noise regime. Thus, the approximate null space of the Hessian recovers the tangent space:

$$\ker(H(\mathbf{x})) \approx T_\mathbf{x}\mathcal{M}. \quad (7)$$

Since $\Pi_{\text{nor}}$ acts as $0$ on the tangent subspace and $1$ on the normal subspace, the leading-order eigenvalues of $H$ are near $0$ and $1/\sigma(t)^2$.

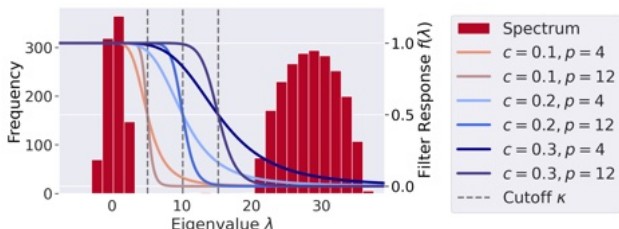

*Figure 2.* LHSD Filter Behavior (Eq. (9)). Filter responses $f(\lambda)$ with varying parameters are overlaid on the Hessian spectrum of $\mathcal{L}^{900} \subset \mathbb{R}^{3072}$ dataset. Increasing $c$ shifts the cutoff $\kappa$ rightward, while increasing $p$ steepens the transition. The cutoff consistently falls within the spectral gap (noise scale $t = 0.04$).

**Property 1** (Tangent–Normal Spectral Separation). *For sufficiently small $\sigma(t)$, the eigenvalues of $H(\mathbf{x})$ near $\mathcal{M}$ separate into two groups:*

- ***Tangent directions*** *(dim $= d$, $|\lambda_i| \approx O(1)$), corresponding to the tangent space $T_\mathbf{x}\mathcal{M}$;*

- ***Normal directions*** *(dim $= D - d$, $\lambda_j \approx 1/\sigma(t)^2 + O(1)$), corresponding to the normal bundle.*

We empirically confirm clear spectral separation for small $\sigma(t)$ (Fig. 2 and App. C). As $\sigma(t)$ increases, the distribution becomes isotropic and the spectral gap collapses; the spectral collapse and its detectability are examined in App. E.

**Definition of LHSD.** Since the tangent space corresponds to the null eigenspace of the Hessian, the intrinsic dimension can be characterized as the multiplicity of zero eigenvalues of $H(\mathbf{x})$. We estimate this quantity by applying a smooth spectral filter $f$ that emphasizes near-zero curvature components:

$$\text{LID}(\mathbf{x}) \approx \text{LHSD}(\mathbf{x}) := \sum_{i=1}^{D} f(\lambda_i) = \text{tr}(f(H(\mathbf{x}))).^{[1]} \quad (8)$$

The equality above is an exact spectral identity: for the symmetric Hessian $H = U\Lambda U^\top$, the matrix function is defined as $f(H) := U f(\Lambda) U^\top$, so $\text{tr}(f(H)) = \sum_{i=1}^{D} f(\lambda_i)$. The later discussion that $f(\lambda)$ is close to 1 or 0 is used to justify the approximation $\text{LID}(\mathbf{x}) \approx \text{LHSD}(\mathbf{x})$, namely that $f$ acts as a soft indicator of tangent versus normal directions. In practice, we further replace the true Hessian by $H_\theta(\mathbf{x}, t) := -\nabla_\mathbf{x} s_\theta(\mathbf{x}, t)$ and approximate $\text{tr}(f(H_\theta))$ numerically via SLQ.

We employ the Hill filter for its *maximally flat passband*, which empirically separates tangent and normal components better than sigmoid-type filters while remaining smooth

---

[1]We omit the dependence of LHSD on $t$ and the score model for simplicity.

enough for accurate SLQ:

$$f(\lambda; \sigma(t)) = \frac{1}{1 + (|\lambda|/\kappa(t))^p}.^{[2]} \quad (9)$$

To adapt to the noise-dependent curvature, we introduce a time-varying cutoff $\kappa(t) := c/\sigma(t)^2$. This cutoff effectively normalizes the curvature by canceling out the noise scaling ($\propto \sigma(t)^{-2}$) inherent in normal directions. Consequently, the filter behaves selectively: $f(\lambda) \approx 1$ for tangent eigenvalues ($\lambda \ll \kappa(t)$), while $f(\lambda) \approx 0$ for normal eigenvalues ($\lambda \gtrsim \kappa(t)$), effectively counting only the intrinsic degrees of freedom. Fig. 2 and App. C illustrate the filter behavior and hyperparameter sensitivity. Here, $c$ shifts the cutoff threshold laterally, and $p$ adjusts the transition steepness.

**Verifiable parameter selection.** LHSD requires the filter cutoff $\kappa(t)$ to reside within the spectral gap. We enable explicit verification of this alignment via a diagnostic indicator. We fix $c$ and $p$ and select $t$. Since varying $t$ shifts both the cutoff and the spectrum simultaneously, a properly selected $t$ secures the cutoff within the spectral gap, allowing tolerance in $c$ and $p$ (Fig. 2). To guide this selection, we use the *transition mass* $M(t)$, defined as

$$M(t) := \frac{1}{D} \sum_{i=1}^{D} \mathbb{I}\left(\lambda_i(t) \in [\kappa(t) - \delta, \kappa(t) + \delta]\right), \quad (10)$$

where $\mathbb{I}(\cdot)$ is the indicator function and $\delta$ denotes the margin (set to $0.2$) around the cutoff. Fig. 3 and App. D illustrate this diagnosis. The "safe zone" (blue bar) is identified as the valley where $M(t) \approx 0$ between spectral peaks. For the filter $f$ defined by $c$ and $p$, the optimal $t$ should be selected from this range.

**Scalable implementation via SLQ.** Direct evaluation of Eq. (8) incurs prohibitive $O(D^3)$ cost. To circumvent this, we employ Stochastic Lanczos Quadrature (SLQ) (Ubaru et al., 2017). We efficiently compute Hessian–vector products $H(\mathbf{x})\mathbf{v} \approx -\nabla(s_\theta(\mathbf{x})^\top \mathbf{v})$ via automatic differentiation, enabling trace estimation by combining Hutchinson's estimator (Hutchinson, 1989) with Lanczos tridiagonalization:

$$\text{tr}(f(H)) \approx \mathbb{E}_\mathbf{v}\left[\mathbf{v}^\top f(H)\mathbf{v}\right] \approx \mathbb{E}_\mathbf{v}\left[\|\mathbf{v}\|^2 \sum_{j=1}^{m} \tau_j^2 f(\tilde{\lambda}_j)\right].$$

See Algorithm 1 and App. B for details. This procedure reduces the complexity to linear time $O(D)$. We find $m = 5$ steps sufficient (Sec. 5).

## 4. Analysis

We analyse the robustness of LHSD against three sources of error: (1) Lanczos approximation error, (2) score approximation error, and (3) Hutchinson variance, followed by

---

[2]Tangent eigenvalues may be slightly negative due to noise or curvature; we use $|\lambda|$ to identify them as tangent components.

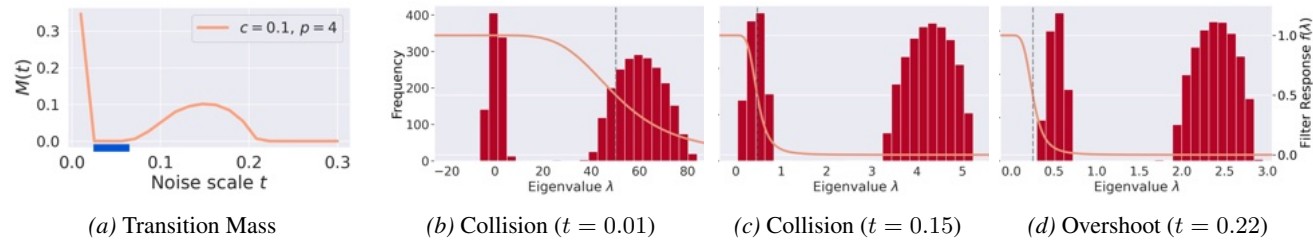

*(a)* Transition Mass      *(b)* Collision ($t = 0.01$)      *(c)* Collision ($t = 0.15$)      *(d)* Overshoot ($t = 0.22$)

*Figure 3.* Selection of $t$ on $\mathcal{L}^{900} \subset \mathbb{R}^{3072}$ for filter with $c = 0.1$ and $p = 4$. (a): Transition mass $M(t)$ identifies a "safe zone" (blue bar): the valley where $M(t) \approx 0$ between two collision peaks. See Fig. 2 for selected safe setting $t = 0.04$. (b)–(d): Cases with $t$ selected outside the safe zone. In (b) and (c), the filter's transition band overlaps with the normal and tangent clusters, respectively. In (d), the cutoff $\kappa$ (gray dashed vertical line) has shifted beyond both clusters; this region is invalid despite $M(t) \approx 0$.

*Table 1.* Mean Absolute Error (MAE), lower is better. $D$ denotes the ambient dimension. The top two results for each column are highlighted in bold. LHSD ($m = 5$) consistently achieves the best performance across high-dimensional and nonlinear settings.

| Method | $D = 100$ | | $D = 1024$ | | | $D = 3072$ | | | | Avg. |
|---|---|---|---|---|---|---|---|---|---|---|
| | $\mathcal{L}^{10+30+90}$ | $\mathcal{F}^{10+25+50}$ | $\mathcal{L}^{900}$ | $\mathcal{L}^{10+80+200}$ | $\mathcal{F}^{10+80+200}$ | $\mathcal{L}^{900}$ | $\mathcal{F}^{900}$ | $\mathcal{L}^{10+80+200}$ | $\mathcal{F}^{10+80+200}$ | |
| ESS | 28.58 | 12.06 | 825.06 | 75.50 | 75.43 | 825.08 | 824.00 | 76.03 | 74.67 | 312.9 |
| LPCA | 31.59 | 17.01 | 891.00 | 75.15 | 77.08 | 891.00 | 891.00 | 77.77 | 73.53 | 336.1 |
| NB | 1.75 | 73.27 | 60.32 | 528.95 | 937.82 | 2171.00 | 2171.00 | 2949.10 | 2985.80 | 1319.9 |
| LIDL | 63.70 | 74.31 | 64.53 | 477.53 | 903.20 | 2175.78 | 2170.88 | 2006.68 | 454.24 | 932.3 |
| FLIPD | **0.76** | 13.73 | **4.26** | 86.03 | 373.80 | **7.78** | 782.50 | 256.40 | 1240.89 | 307.4 |
| LHSD ($m = 2$) | **1.59** | **7.66** | **4.77** | **8.97** | **21.30** | 35.00 | **39.96** | **37.64** | **29.70** | 20.7 |
| LHSD ($m = 5$) | 1.64 | **2.16** | 5.03 | **3.47** | **6.90** | **11.53** | **18.79** | **4.70** | **5.19** | 6.6 |

*Table 2.* MAE comparison in low-dimensional space ($D = 3$).

| Method | Moon | Funnel | $\mathcal{L}^{1+2+3}$ | Avg. |
|---|---|---|---|---|
| NB | 0.708 | 0.505 | 0.294 | 0.502 |
| LIDL | 0.251 | 0.626 | 0.297 | 0.391 |
| FLIPD | 0.456 | 0.363 | 0.219 | 0.346 |
| LHSD ($m = 2$) | **0.228** | **0.323** | **0.195** | **0.249** |

*Table 3.* MAE comparison on IDR data ($D = 784$).

| Method | Moon | Funnel | $\mathcal{L}^{1+2+3}$ | Avg. |
|---|---|---|---|---|
| NB | 1.014 | 2.308 | 20.55 | 3.698 |
| LIDL | 97.87 | 113.00 | 104.02 | 45.21 |
| FLIPD | **0.284** | 1.606 | 1.637 | 0.702 |
| LHSD ($m = 2$) | 0.691 | **0.661** | **0.629** | **0.425** |

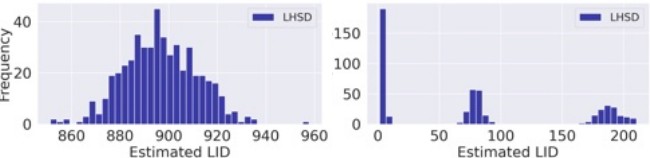

*Figure 4.* LID estimation by LHSD on (Left) $\mathcal{L}^{900} \subset \mathbb{R}^{3072}$ with MAE 11.53, and (Right) $\mathcal{F}^{10+80+200} \subset \mathbb{R}^{3072}$ with MAE 5.19.

an analysis of its computational efficiency. The robustness of LHSD is underpinned by the TangentNormal Separation (Property 1), serving as a crucial mechanism for suppressing various estimation errors.

**Robustness to Lanczos Approximation Error ($m = 5$).** Although the quadrature error $|E_{\text{Lanczos}}| \propto \frac{|f^{(2m)}(\xi)|}{(2m)!}$ typically necessitates a large rank $m$, LHSD is robust even at $m = 5$ due to Property 1. Since high-order derivatives $f^{(2m)}$ vanish in the filter's flat saturation regions, placing the cutoff $\kappa$ within the spectral gap (Fig. 2, App. C) effectively nullifies the error term. This yields high accuracy even at $m = 2$ (Sec. 5). To further ensure stability, we use a moderate steepness $p = 4$ to limit derivative magnitudes in the transition band.

**Robustness to Score Approximation Error.** Trace-based estimators (FLIPD and LIDL) are vulnerable to the score approximation error $\mathbf{e}$ in the learned score $\hat{\mathbf{s}} = \mathbf{s} + \mathbf{e}$, which induces a Hessian error $E = -\nabla \mathbf{e}$ in $\hat{H} = H + E$. They directly use the trace $\text{tr}(\hat{H})$, which decomposes as $\text{tr}(H) + \text{tr}(E)$. Under Property 1, the true Hessian eigenvalues scale as $\lambda_i^{\text{tan}} \approx 0$ for $i \leq d$ and $\lambda_j^{\text{nor}} \approx \alpha \, \sigma(t)^{-2}$ for $j > d$, so that $\text{tr}(H) \approx (D - d) \, \alpha \, \sigma(t)^{-2}$. This implies that the error term $\text{tr}(E)$ is also an indiscriminate sum over all directions. Even if the per-direction score error is small, its normal components accumulate across the high co-dimension ($D - d$), causing $\text{tr}(E)$ to scale proportionally to the ambient dimension $D$. As a result, trace-based estimators suffer from a severely degraded S/N ratio in high-dimensional and small-noise regimes.

In contrast, LHSD estimates $\text{tr}(f(\hat{H}))$. A first-order Taylor expansion yields the approximation as $\text{tr}(f(\hat{H})) - \text{tr}(f(H)) \approx \text{tr}(f'(H)E)$. Just as with the higher-order derivatives in Lanczos error, the first derivative $f'(\lambda)$ takes non-zero values only in the vicinity of the cutoff $\kappa$. Pro-

vided that the spectral separation holds and the cutoff $\kappa$ lies within the spectral gap (Fig. 2), the derivative $f'(\lambda)$ is negligible for all eigenvalues. Consequently, the sensitivity term $\text{tr}(f'(H)E)$ effectively vanishes. This mechanism suppresses the influence of the error $E$ regardless of the ambient dimension, rendering LHSD robust even in high-dimensional settings (Experiment 4).

**Robustness to Hutchinson Variance ($K = 8$).** We analyze the stochastic variance governing the probe count $K$ in Hutchinson's estimator. Trace-based estimators (FLIPD and LIDL) suffer from instability because Hutchinson variance scales with the Frobenius norm $\text{Var}[v^\top Hv] \propto \|H\|_F^2$. Under Property 1, this norm is dominated by normal curvature: $\|H\|_F^2 \approx \sum_{j>d}(\lambda_j^{\text{nor}})^2 \approx (D-d)\alpha^2\sigma(t)^{-4}$. This causes variance explosion in high co-dimensions, explaining the degradation of stochastic FLIPD in Experiment 3 (Sec. 5). In contrast, LHSD operates on the filtered operator $f(H)$, where the variance is $\text{Var}[\mathbf{v}^\top f(H)\mathbf{v}] = 2\|f(H)\|_F^2 = 2\sum_i(f(\lambda_i))^2$.[3] Since the filter maps tangent eigenvalues to $\approx 1$ and normal ones to $\approx 0$, this sum reduces to $\approx 2d$. Thus, variance depends only on the intrinsic dimension $d$, not the ambient dimension $D$, enabling stable estimation with minimal probes ($K = 8$).

**Computational Complexity.** We compare the computational cost. Thanks to the robustness detailed above, LHSD requires only a few Lanczos steps ($m = 5$) and probes ($K = 8$). Let $C_{\text{HVP}}$ denote the cost of one Hessian-vector product, which is $\mathcal{O}(D)$.

*LHSD (Ours):* The total cost is $T_{\text{LHSD}}(\mathbf{x}) = mK \cdot C_{\text{HVP}}$. The dependence on dimension $D$ is solely through $C_{\text{HVP}}$. This ensures strictly linear scaling $\mathcal{O}(D)$, making LHSD scalable to high-dimensional data.

*FLIPD & LIDL:* The bottleneck is computing the Laplacian $\text{tr}(\nabla \mathbf{s}_\theta)$. Exact computation requires $D$ backpropagations, costing $T_{\text{FLIPD}}(\mathbf{x}) \approx D \cdot C_{\text{HVP}}$, which scales quadratically as $\mathcal{O}(D^2)$. LIDL further multiplies this by $L$ noise scales. For a $32 \times 32$ image ($D = 3072$), while LHSD requires only 16 HVPs, FLIPD requires 3072 and LIDL tens of thousands, resulting in an overwhelming difference in computational cost (see Sec. 5).

*NB:* NB requires SVD on $M \approx 4D$ score vectors. The cost $T_{\text{NB}}(\mathbf{x}) \approx MC_{\text{score}} + \mathcal{O}(DM^2)$ scales as $\mathcal{O}(D^3)$, rendering it intractable for high-dimensional data.

In summary, the computational complexity follows LHSD $\ll$ FLIPD $<$ LIDL $\ll$ NB. LHSD is the only method scaling linearly with $D$, enabling fast inference even for dimensions exceeding 3000.

---

[3]With Rademacher probing, the variance is further reduced: $\text{Var}[\mathbf{v}^\top f(H)\mathbf{v}] = 2\|f(H)\|_F^2 - 2\|\text{diag}(f(H))\|_2^2$.

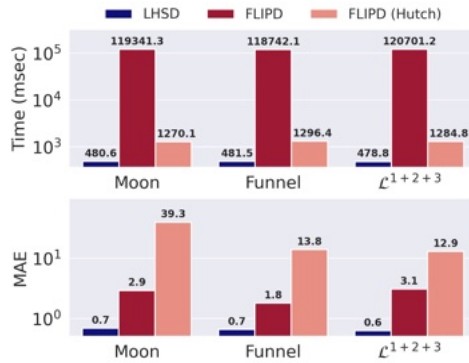

*Figure 5.* Computational efficiency (top) and estimation error (bottom) on IDR ($D = 784$). LHSD ($m = 2$) achieves a $>250\times$ speedup over FLIPD. While the stochastic variant (FLIPD-Hutch) reduces computational cost, it incurs severe accuracy degradation. In contrast, LHSD demonstrates superior performance, simultaneously achieving high efficiency and low estimation error.

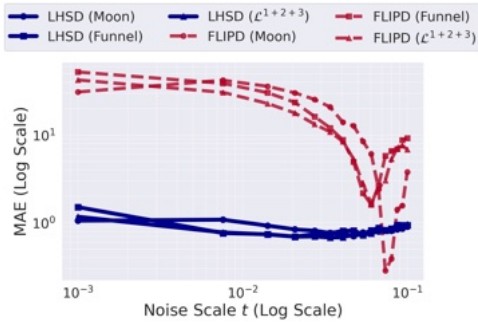

*Figure 6.* Robustness to noise scale $t$. LHSD (blue) maintains consistently low estimation error (MAE) across a wide range of noise scales on all datasets.

## 5. Experiments

We evaluate LHSD on synthetic manifolds where the ground-truth LID $d$ is known. We compare against diffusion-based (FLIPD, LIDL, NB) and $k$NN-based (ESS, LPCA) methods. For fairness, all diffusion methods use identical pre-trained models (UNet2D for ambient dimension $D \geq 3$, MLP for $D = 3$) and the same VP-SDE (see App. F for the model specification). For Experiments 1 and 2, the model was trained for 500 epochs on 500k samples, and evaluated on 2,000 samples. For baselines, we employ the official code from Kamkari et al. (2024b) (diffusion-based) and `scikit-dimension` (Bac et al., 2021) ($k$NN-based).

### Experiment 1: High-Dimensional Manifolds

Following Kamkari et al. (2024b), we generated mixtures of disjoint sub-manifolds embedded in $\mathbb{R}^D$ (linearly $\mathcal{L}$ or non-linearly $\mathcal{F}$). The superscript notation denotes the intrinsic dimensions of the components; for example, $\mathcal{F}^{10+80+200}$ consists of three sub-manifolds with dimensions $10, 80$, and $200$. We sampled $N = 2,000$ points equiprobably from

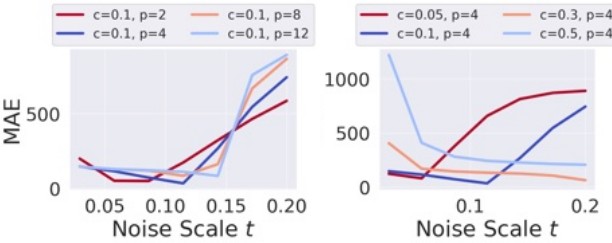

*Figure 7.* Impact of filter parameters $p$ and $c$. Left: Variation in steepness $p$. Right: Variation in cutoff $c$. Performance is largely stable across $p$ values, but shows sensitivity to $c$, where extreme values may lead to degradation. Evaluated on $\mathcal{L}^{900} \subset \mathbb{R}^{3072}$.

*Table 4.* Ablation on score-model quality for the 1024-dimensional 10+80+200 manifold mixture.

| Setting | Train samples | Training loss | MAE |
|---|---|---|---|
| $\mathcal{L}^{10+80+200}$ | 500k | 57.51 | 3.47 |
| | 50k | 70.80 | 4.16 |
| $\mathcal{F}^{10+80+200}$ | 500k | 135.63 | 6.90 |
| | 50k | 140.00 | 7.52 |

these components. The ground-truth LID $d^*(\mathbf{x})$ varies per sample depending on its originating component. Performance is evaluated using the Mean Absolute Error (MAE) averaged over all samples: $\frac{1}{N}\sum_i |\hat{d}(\mathbf{x}_i) - d^*(\mathbf{x}_i)|$. We varied: (1) ambient dimension ($D \in [100, 3072]$), (2) dimensional gap ($d \ll D$ vs. $d \approx D$), and (3) embedding nonlinearity ($\mathcal{L}$ vs. $\mathcal{F}$). For LHSD, we fixed $c = 0.1, p = 4$ and selected $t$ via the transition mass diagnostic. For baselines (FLIPD, NB) lacking such diagnostics, we report the best performance over a range of $t \in [0.001, 0.2]$.

**Results** (Table 1). While increasing $D$ and nonlinearity complicates estimation, LHSD minimizes their impact. Fig. 4 and App. I confirm that LHSD accurately recovers intrinsic distributions. While increasing $D$ and nonlinearity complicates estimation, LHSD minimizes their impact, accurately recovering intrinsic distributions (App. I). LHSD maintains consistently low MAE across all settings from 100 to 3072 dimensions, achieving an average error approximately $1/46$ that of the second-best method, FLIPD, with $m = 5$ (and $1/14$ even with $m = 2$). Evaluations at $m = 10, 20$ yielded negligible gains over $m = 5$ despite the linear increase in computational cost; hence, $m = 5$ is optimal (see App. H).

**Robustness in High-Dimensional Regimes.** In high co-dimension settings ($d \ll D$) typical of real-world data, FLIPD degrades severely, yielding an MAE of 1240.89 on $\mathcal{F}^{10+80+200} \subset \mathbb{R}^{3072}$; in contrast, LHSD achieves 5.19 (a reduction of $\approx 1/239$). Similarly, at high intrinsic dimensions ($d = 900$), FLIPD collapses under nonlinearity ($\mathcal{F}^{900}$: 782.5) despite performing well on linear data, whereas LHSD remains robust (18.79). These failures stem from FLIPD's vulnerability to errors (noise or score ap-

proximation) accumulating across the vast normal subspace ($D - d$), as analyzed in Sec. 4. LHSD successfully masks these errors via spectral filtering, ensuring robust estimation regardless of co-dimension or nonlinearity.

**Failure of $k$NN-based Methods.** Meanwhile, $k$NN-based methods, ESS and LPCA, failed as the intrinsic dimension $d$ increased, regardless of the ambient dimension size. We set the neighbor count to $n$=200; increasing $n$ further yielded no improvement. This indicates an intrinsic failure: the curse of dimensionality fundamentally compromises the reliability of nearest neighbor search in high-dimensional manifolds.

### Experiment 2: IDR Benchmarks

We further evaluate robustness using the Intrinsic Dimension Recovery (IDR) benchmark (Tempczyk et al., 2026). From the benchmark suite, we select the Moon and Funnel, and additionally introduce $\mathcal{L}^{1+2+3}$ (Fig. 1). The ground-truth LID $d^*(\mathbf{x})$ is defined based on the manifold geometry (see App. G.3 for definitions). We assess performance in two settings: (1) the original 3D space ($D = 3$), and (2) a high-dimensional embedding ($D = 784$) obtained by nonlinearly mapping the manifolds into the Fashion-MNIST space (see App. G.2). Previous studies reported that existing diffusion-based methods fail significantly in this high-dimensional embedding.

**Results** (Tables 2 & 3). Consistent with the previous study, we confirmed that FLIPD, NB, and LIDL suffer dramatic performance degradation due to the IDR embedding (e.g., FLIPD MAE on Funnel: $0.363 \rightarrow 1.606$). Conversely, LHSD maintained stable behavior with minimal MAE degradation (e.g., Funnel: $0.323 \rightarrow 0.661$), even with a minimal Lanczos rank $m = 2$ (matching the performance at $m = 5$). On this dataset, LHSD showed robust performance.

### Experiment 3: Efficiency and Parameter Sensitivity

**Computational efficiency.** We evaluated the computational efficiency and robustness to Hutchinson approximation (see App. J for the measurement protocol). Fig. 5 shows the results on the IDR benchmarks. Even FLIPD, the second fastest among diffusion-based methods, requires approximately 120,000 ms for inference. In contrast, LHSD ($m = 2$) completes the task in $\approx 480$ ms, achieving over a $250\times$ speedup. Increasing to $m = 5$ raised the runtime to $\approx 1,200$ ms with negligible performance gains. When applying Hutchinson ($K = 8$) to FLIPD for fair comparison, runtime dropped to $\approx 1,280$ ms, but error spiked catastrophically (MAE $>12.9$). Conversely, LHSD maintained high precision (MAE $\approx 0.6$) even with $m = 2$. As discussed in Sec. 4, this is because the spectral filter effectively suppresses the variance, enabling stable estimation with few probes.

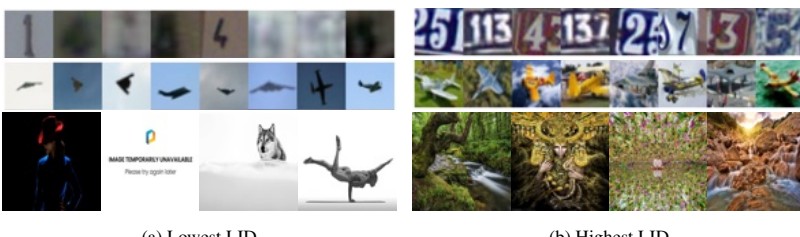

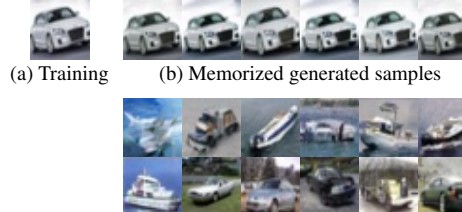

(a) Training      (b) Memorized generated samples

(a) Lowest LID            (b) Highest LID

(c) Non-memorized generated samples

*Figure 8.* Comparison of generated samples with the lowest and highest LID values estimated by LHSD. SVHN, CIFAR-10 ('airplane' class), and LAION-Aesthetics are shown from top to bottom. LHSD successfully distinguishes between visually simple images (e.g., flat backgrounds, single objects) and complex ones (e.g., dense textures, cluttered scenes) across different resolutions and domains.

*Figure 9.* DDPM samples on CIFAR-10. (b) Memorized near-duplicates of training sample (a). (c) Non-memorized samples selected for similar complexity to (b).

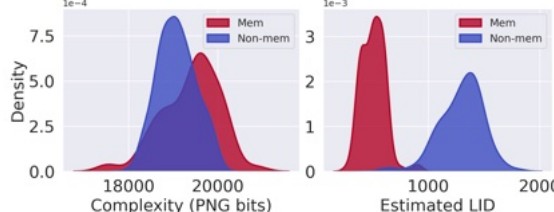

*Figure 10.* The PNG complexity distributions (left) for the Memorized and Non-memorized sets overlap significantly, ruling out image complexity as a confounding factor. Under this controlled condition, the Estimated LID (right) shows a distinct separation, demonstrating that LHSD can detect memorized samples by capturing their low-intrinsic dimensionality.

**Sensitivity to noise scale $t$.** As shown in Fig. 6, LHSD remains robust across a wide range of $t$ on IDR benchmarks ($D = 784$); see App. K for visualizations of the 3D reconstructions. Although sensitivity increases in higher dimensions ($D > 1000$; see App. H, Fig. 14), the *transition mass* diagnostic proves decisive, enabling users to select the optimal $t$. In contrast, FLIPD exhibits two critical flaws: it permits theoretically invalid negative estimates (Figs. 17–23) and lacks a mechanism to select the optimal $t$ in unsupervised settings, unlike LHSD.

**Sensitivity to filter parameters.** We investigated the impact of parameters $c$ and $p$ on MAE using the $\mathcal{L}^{900} \subset \mathbb{R}^{3072}$ dataset (consistent with Fig. 2). Fig. 7 indicates that performance is relatively insensitive to $p$, while $c$ has a moderate influence. Crucially, however, the validity of the selected parameters is explicitly diagnosable in practice; users can verify the configuration by checking the transition mass and overlaying the filter profile on the spectral distribution.

### Experiment 4: Ablation on Score Model Quality

As an ablation study, we examined how score-model quality affects LHSD estimation by reducing the score-model training set from 500k to 50k samples in the $D = 1024$ $\mathcal{L}^{10+80+200}$ and $\mathcal{F}^{10+80+200}$ settings. The results are sum-

marized in Table 4. For all experiments, the score model was trained for 500 epochs, and LHSD was evaluated with $m = 5$, $c = 0.1$, and $p = 4$. As expected, reducing the training data worsened the training loss. However, the downstream LHSD estimation error increased only modestly (Linear: $3.47 \rightarrow 4.16$; Nonlinear: $6.90 \rightarrow 7.52$), suggesting that, in these settings, LHSD is reasonably robust to moderate score-quality degradation.

### Experiment 5: LID Estimation on Images

To demonstrate LHSD's capability to capture visual complexity, we utilized SVHN, CIFAR-10, and LAION-Aesthetics (Netzer et al., 2011; Krizhevsky et al., 2009; Schuhmann et al., 2022). While we used standard UNet2D models for SVHN and CIFAR-10, we employed Stable Diffusion v1.5 for the high-resolution ($512 \times 512$) LAION-Aesthetics dataset (Rombach et al., 2022).[4] For the latter, LID was estimated in the variational autoencoders space ($D \approx 1.6 \times 10^4$) rather than pixel space, following the premise in (Kamkari et al., 2024b) that the encoder's continuity preserves the data topology.

**Results.** Fig. 8 presents the results of sorting images based on their estimated LID. We observed that images with low LID possess uniform backgrounds or smooth color gradients, whereas images with high LID tend to be filled with fine textures, cluttered objects, and high-frequency details. This aligns with the intuition that image complexity correlates with local dimension, suggesting that LHSD quantifies this degree in a manner consistent with human perception, even in pixel and extremely high-dimensional latent spaces.

### Experiment 6: Memorization Detection

Previous studies suggest that memorized samples in diffusion models possess anomalously low LID. Based on this insight, we investigate whether LHSD can detect memorization using an official pre-trained DDPM on CIFAR-10.[5] We

---

[4]huggingface.co/stable-diffusion-v1-5/stable-diffusion-v1-5
[5]huggingface.co/docs/diffusers/api/pipelines/ddpm

constructed a set of memorized images ($\mathcal{S}_{\text{mem}}$) and a comparative non-memorized set ($\mathcal{S}_{\text{non}}$) following the protocol of Ross et al. (2025) (see App. L for construction details). To ensure LHSD is not merely reacting to image simplicity, we controlled for *visual complexity*; as shown in Fig. 10 (left), the PNG complexity distributions of $\mathcal{S}_{\text{mem}}$ and $\mathcal{S}_{\text{non}}$ are adjusted to overlap.

**Results.** Under these conditions, the LID of $\mathcal{S}_{\text{mem}}$ was clearly separated from that of $\mathcal{S}_{\text{non}}$ (Fig. 10 right), showing distinctively lower values. It achieved a high AUROC of 99.87% and a similarly high AUPR of 99.55%. These results demonstrate that LHSD captures the low-dimensional structure specific to memorization, independent of image complexity, supporting its effectiveness as an LID-based detection method.

## 6. Conclusion and Limitations

We proposed Local Hessian Spectral Dimension (LHSD), which recovers intrinsic dimensionality from the spectral geometry of the log-density Hessian. Leveraging SLQ for scalable computation, LHSD suppresses normal-direction curvature and remains accurate in high-dimensional regimes where existing diffusion-based methods degrade.

A fundamental advantage of LHSD over prior approaches is its *verifiability*. By overlaying the filter profile on the Hessian eigenvalue distribution, practitioners can visually assess if the transition band lie within the spectral gap. This enables practitioners to directly validate parameter choices $(c, p, t)$ and diagnose whether the underlying manifold structure is reliably captured, providing an interpretability and reliability mechanism absent in previous methods.

A limitation of LHSD is its reliance on the quality of the trained score model. While we assessed training fidelity via loss convergence and distributional overlap between generated and training samples, it remains unclear whether such criteria ensure accurate recovery of the true Hessian spectral geometry, particularly for complex real-world data. Establishing rigorous diagnostics for spectral consistency of score models is an important open problem.

## Impact Statement

LHSD is a methodological tool for analyzing low-dimensional structure in high-dimensional data. Its primary positive impact is to improve geometric diagnostics for modern machine learning models, including the analysis of robustness and generative-model behavior. More broadly, local dimension and manifold analysis are also relevant in scientific domains where high-dimensional observations are believed to lie near lower-dimensional structure, such as cell-type discovery in single-cell genomics and neural pop-

ulation analysis in neuroscience. Better local geometric estimators may therefore benefit both safer machine learning research and data-driven scientific discovery. However, such estimates may be over-interpreted if applied without sufficient validation, since they depend on modeling assumptions and approximation quality. For this reason, careful empirical evaluation and transparent reporting of limitations are important for responsible downstream use.

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

# A. Score-based Diffusion Models

Following the framework of Song et al. [2021], we describe the data distribution as a continuous-time stochastic differential equation (SDE).

**Forward Diffusion Process.** For a $D$-dimensional data point $\mathbf{x}(0) \in \mathbb{R}^D$ following the data distribution $p_0(\mathbf{x})$, the forward process, which continuously adds noise over time $t \in [0, T]$, is defined by the following Itô SDE:

$$d\mathbf{x} = \mathbf{f}(\mathbf{x}, t)dt + g(t)d\mathbf{w} \tag{11}$$

where $\mathbf{f}(\cdot, t) : \mathbb{R}^D \to \mathbb{R}^D$ is the drift coefficient, $g(t) \in \mathbb{R}$ is the diffusion coefficient, and $\mathbf{w}$ is the standard Wiener process. A key property of this SDE is that the transition kernel $p_{0t}(\mathbf{x}(t)|\mathbf{x}(0))$ given an initial value $\mathbf{x}(0)$ becomes a Gaussian distribution:

$$p_{0t}(\mathbf{x}(t)|\mathbf{x}(0)) = \mathcal{N}(\mathbf{x}(t); \mu(t)\mathbf{x}(0), \sigma^2(t)\mathbf{I}) \tag{12}$$

Here, $\mu(t)$ and $\sigma(t)$ are mean and variance parameters uniquely determined by the SDE coefficients $\mathbf{f}$ and $g$.

**Reverse Diffusion Process.** The generation process is realized by reversing this diffusion process in time. According to the theory by Anderson [1982], the reverse-time diffusion process also follows an SDE:

$$d\mathbf{x} = [\mathbf{f}(\mathbf{x}, t) - g(t)^2 \nabla_{\mathbf{x}} \log p_t(\mathbf{x})]dt + g(t)d\bar{\mathbf{w}} \tag{13}$$

where $\bar{\mathbf{w}}$ denotes the standard Wiener process in reverse time. Simulating this reverse process requires the score function $\nabla_{\mathbf{x}} \log p_t(\mathbf{x})$ at each time step $t$.

**Score Estimation via Denoising Score Matching.** Since the true score function is unknown, we approximate it using a time-dependent neural network (score model) $\mathbf{s}_\theta(\mathbf{x}, t)$. For training, we use the continuous-time extension of the Denoising Score Matching objective [Vincent, 2011]:

$$\mathcal{L}(\theta) = \mathbb{E}_{t, \mathbf{x}(0), \mathbf{x}(t)} \left[ \lambda(t) \| \mathbf{s}_\theta(\mathbf{x}(t), t) - \nabla_{\mathbf{x}(t)} \log p_{0t}(\mathbf{x}(t)|\mathbf{x}(0)) \|_2^2 \right] \tag{14}$$

where $\lambda(t)$ is a weighting function. Since the transition kernel $p_{0t}$ is Gaussian, the target conditional score $\nabla_{\mathbf{x}(t)} \log p_{0t}(\mathbf{x}(t)|\mathbf{x}(0))$ can be derived analytically:

$$\nabla_{\mathbf{x}(t)} \log p_{0t}(\mathbf{x}(t)|\mathbf{x}(0)) = -\frac{\mathbf{x}(t) - \mu(t)\mathbf{x}(0)}{\sigma^2(t)} \tag{15}$$

Optimization yields $\mathbf{s}_{\theta^*}(\mathbf{x}, t) \approx \nabla_{\mathbf{x}} \log p_t(\mathbf{x})$.

**VP-SDE Configuration.** In our experiments, we adopted the Variance Preserving (VP) SDE, which corresponds to the continuous limit of Denoising Diffusion Probabilistic Models (DDPM) [Ho et al., 2020]. The drift and diffusion coefficients are defined as:

$$\mathbf{f}(\mathbf{x}, t) = -\frac{1}{2}\beta(t)\mathbf{x}, \quad g(t) = \sqrt{\beta(t)} \tag{16}$$

where $\beta(t)$ is the noise schedule function. We implemented the following linear schedule:

$$\beta(t) = \beta_{\min} + t(\beta_{\max} - \beta_{\min}), \quad t \in [0, 1] \tag{17}$$

We set $\beta_{\min} = 0.1$ and $\beta_{\max} = 20.0$. Under this schedule, the mean and variance parameters of the transition kernel are given by:

$$\mu(t) = \exp\left(-\frac{1}{2}\int_0^t \beta(s)ds\right), \quad \sigma^2(t) = 1 - \mu(t)^2. \tag{18}$$

## B. Stochastic Lanczos Quadrature

Stochastic Lanczos Quadrature (SLQ) serves as the core component of our proposed method. We formulate the problem of estimating the spectral distribution of the Hessian $H$ as an integral approximation problem using Gaussian quadrature.

### Trace Estimation as Spectral Integration.

Using Hutchinson's estimator, the computation of the trace is reduced to calculating the expected value of a quadratic form with respect to a random vector $\mathbf{v}$.

$$\text{tr}(f(H)) = \mathbb{E}_{\mathbf{v}}[\mathbf{v}^\top f(H)\mathbf{v}] \tag{19}$$

Given the eigenvalue decomposition $H = U\Lambda U^\top$, this quadratic form can be expressed as a Riemann-Stieltjes integral:

$$\mathbf{v}^\top f(H)\mathbf{v} = \sum_{i=1}^{D} f(\lambda_i)(\mathbf{u}_i^\top \mathbf{v})^2 = \int_{\lambda_{\min}}^{\lambda_{\max}} f(\lambda)d\mu(\lambda) \tag{20}$$

Here, $\mu(\lambda)$ represents a discrete spectral measure associated with the weights $(\mathbf{u}_i^\top \mathbf{v})^2$. The objective of SLQ is to approximate this integral with high precision using a small number of nodes (Lanczos steps $m$).

### Lanczos Algorithm and Moment Matching.

The Lanczos algorithm iteratively generates an orthonormal basis $Q_m$ for the Krylov subspace $\mathcal{K}_m(H, \mathbf{v}) = \text{span}\{\mathbf{v}, H\mathbf{v}, \ldots, H^{m-1}\mathbf{v}\}$ starting from $\mathbf{v}$, while simultaneously constructing a projected tridiagonal matrix $T_m = Q_m^\top H Q_m$.

$$T_m = \begin{pmatrix} \alpha_1 & \beta_2 & & 0 \\ \beta_2 & \alpha_2 & \ddots & \\ & \ddots & \ddots & \beta_m \\ 0 & & \beta_m & \alpha_m \end{pmatrix} \tag{21}$$

Intuitively, this process acts as a spectral compression of the high-dimensional Hessian $H$ ($D \times D$) into a low-dimensional tridiagonal matrix $T_m$ ($m \times m$). Mathematically, this constitutes a Rayleigh-Ritz projection onto the dynamically constructed Krylov subspace, where the eigenvalues of $T_m$ (Ritz values) optimally approximate the spectral components of $H$ along the directions of maximum variation. Despite this drastic reduction in size ($m \ll D$), $T_m$ preserves a critical property known as Moment Matching with respect to the original matrix $H$. Specifically, the first $2m - 1$ moments match exactly:

$$\mathbf{v}^\top H^k \mathbf{v} = \|\mathbf{v}\|^2 \mathbf{e}_1^\top T_m^k \mathbf{e}_1, \quad \forall k \in \{0, \ldots, 2m-1\} \tag{22}$$

This implies that if the filter function $f(\lambda)$ can be well-approximated by a polynomial of degree $2m - 1$ or less, the result computed using $T_m$ will coincide with the true value derived from $H$. Since the sigmoid-like filter $f$ used in LHSD is a smooth function, a low-degree polynomial approximation is effective. This provides the theoretical rationale for achieving high accuracy with a small number of steps $m$.

### Connection to Gaussian Quadrature.

The moment matching property described above is equivalent to the theory of Gaussian Quadrature. Let the eigenvalue decomposition of the tridiagonal matrix be $T_m = Y\tilde{\Lambda}Y^\top$, where $\tilde{\lambda}_j$ denote the Ritz values (approximate eigenvalues) and $\tau_j = Y_{1,j}$ denote the corresponding weights (the first component of the eigenvectors). The approximation of the quadratic form is then given by:

$$\mathbf{v}^\top f(H)\mathbf{v} \approx \|\mathbf{v}\|^2 \mathbf{e}_1^\top f(T_m)\mathbf{e}_1 = \|\mathbf{v}\|^2 \sum_{j=1}^{m} \tau_j^2 f(\tilde{\lambda}_j) \tag{23}$$

This equation is precisely the formula for $m$-point Gaussian quadrature over the spectral interval. In other words, SLQ can be interpreted as adaptively computing the optimal quadrature nodes (Ritz values) and weights for each random starting vector $\mathbf{v}$ to perform spectral integration.

## C. Empirical Verification of Spectral Separation

We provide empirical evidence supporting Property 1 across various datasets. Fig. 11 displays the Hessian eigenvalue distributions for nine different datasets ranging from $D = 100$ to $3072$ used in Sec. 5. In all cases, with an appropriately selected noise scale $t$, the spectrum clearly separates into a tangent cluster ($\lambda \approx 0$) and a normal cluster ($\lambda > 0$). The figure also overlays the profiles of the filter function and their respective cutoff thresholds for the parameter configurations. These results confirm that the tangent-normal separation (Property 1) is consistently observed across the datasets used in our experiments, with the filter cutoff $\kappa$ correctly positioned within the spectral gap. We note that adjusting the noise scale $t$ modifies both the location of the normal cluster and the position of the transition band centered at the filter cutoff. For further details on this mechanism, please refer to App. D.

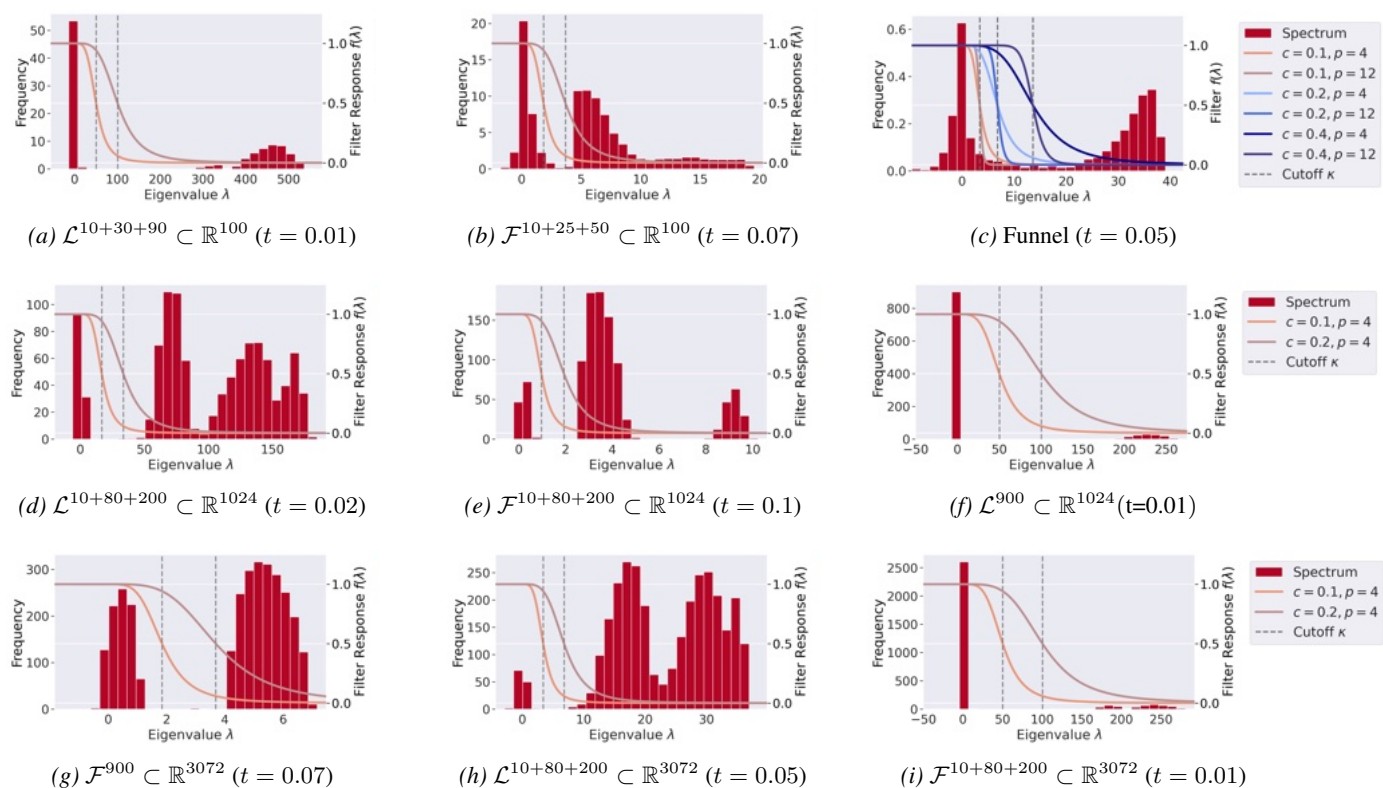

*Figure 11.* Visualization of Hessian spectral separation computed on 20 randomly sampled data points for each dataset. In all shown cases, the noise scale $t$ is chosen such that the filter cutoff $\kappa$ (dashed line) is correctly positioned within the spectral gap between tangent and normal components.

## D. Noise-Scale Selection via Transition Mass

The accuracy of LHSD relies on the alignment between the spectral filter's transition band and the spectral gap of the Hessian. This alignment can be fine-tuned by adjusting the noise scale $t$. Varying $t$ affects the system in two ways: it shifts the filter cutoff $\kappa(t) = c/\sigma(t)^2$ and simultaneously transforms the eigenvalue distribution. Specifically, as the noise scale increases (larger $t$), the normal cluster shifts toward smaller eigenvalues because the stronger Gaussian noise smooths the manifold structure, reducing the curvature in normal directions. To identify a safe range of $t$ that avoids collision between the filter cutoff and the spectral clusters, we use a diagnostic metric called *transition mass $M(t)$* (Sec. 3).

The safe range for $t$ is the interval where $M(t)$ remains close to zero, preceding the onset of significant mass accumulation (the blue bar in the figure). Analogous to the case of $\mathcal{L}^{900} \subset \mathbb{R}^{3072}$ in the main text, we here demonstrate the selection of $t$ via $M(t)$ for $\mathcal{F}^{10+25+50} \subset \mathbb{R}^{100}$ and $\mathcal{F}^{10+80+200} \subset \mathbb{R}^{1024}$, shown in Fig. 12. The filter parameters were set to $c = 0.1$ and $p = 4$.

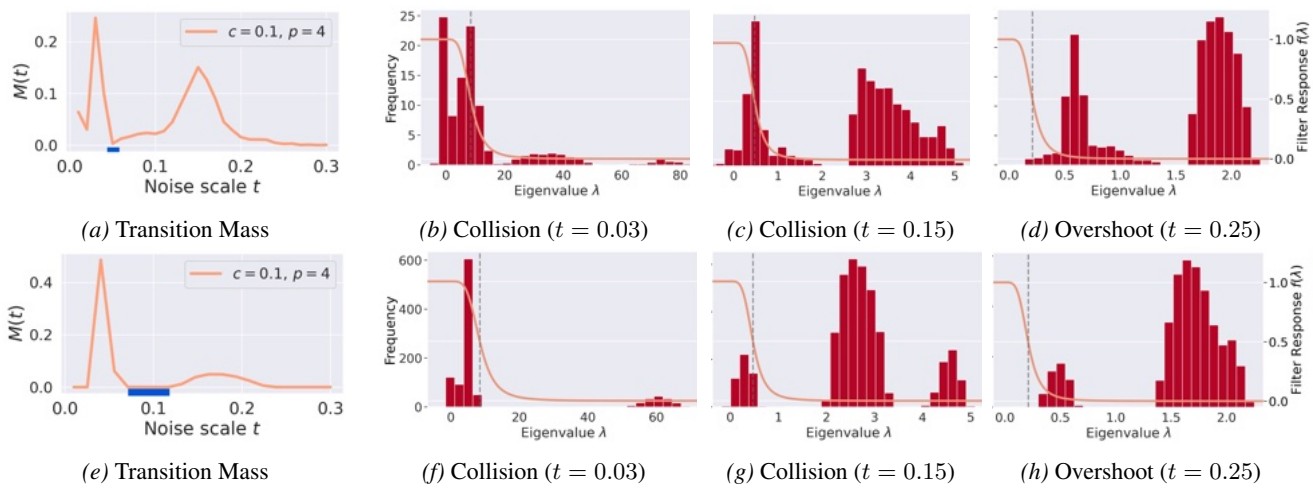

*Figure 12.* Selection of $t$ on $\mathcal{F}^{10+25+50} \subset \mathbb{R}^{100}$ (Top) and $\mathcal{F}^{10+80+200} \subset \mathbb{R}^{1024}$ (Bottom). (a),(e): Transition mass $M(t)$ identifies a "safe zone" (blue bar). (b)–(d) and (f)–(h): Cases with $t$ selected outside the safe zone. See Figs. 11b and 11e for selected safe setting.

## E. Spectral Gap and Collapse

A prerequisite for LHSD is the existence of a spectral gap separating the tangent and normal eigenvalues. We empirically verified that a distinct spectral gap persists across all datasets used in our experiments within the valid range of noise scales (App. C). However, this separation naturally degrades under excessive noise. As the noise scale $t$ increases, two effects occur simultaneously: (1) The normal eigenvalues ($\lambda \approx 1/\sigma(t)^2$) decrease and shift toward zero. (2) The tangent eigenvalues spread outward as the diffusion destroys the local manifold structure. Eventually, these two clusters merge, leading to *spectral collapse*, where tangent and normal components become indistinguishable.

Fig. 13 illustrates this phenomenon on the $\mathcal{F}^{900} \subset \mathbb{R}^{3072}$ and $\mathcal{F}^{10+25+50} \subset \mathbb{R}^{100}$ datasets. For $\mathcal{F}^{900} \subset \mathbb{R}^{3072}$, a distinct spectral gap persists at $t = 0.60$, allowing the filter to isolate tangent components, whereas at $t = 0.65$, the clusters collide, closing the spectral gap. Similarly, for $\mathcal{F}^{10+25+50} \subset \mathbb{R}^{100}$, the separation holds at $t = 0.5$ but collapses at $t = 0.6$.

Importantly, the diagnostic $M(t)$ reveals that the safe zone (spectral valley) resides at smaller $t$ (e.g., as shown in App. D, Fig. 12a). Thus, it is evident from the diagnostic that such large $t$ falls outside the safe zone.

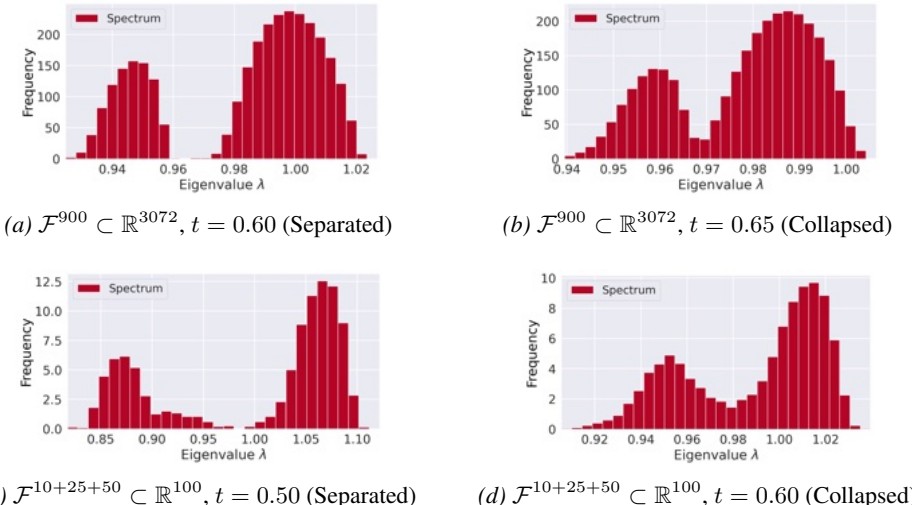

*(a)* $\mathcal{F}^{900} \subset \mathbb{R}^{3072}$, $t = 0.60$ (Separated)  *(b)* $\mathcal{F}^{900} \subset \mathbb{R}^{3072}$, $t = 0.65$ (Collapsed)

*(c)* $\mathcal{F}^{10+25+50} \subset \mathbb{R}^{100}$, $t = 0.50$ (Separated)  *(d)* $\mathcal{F}^{10+25+50} \subset \mathbb{R}^{100}$, $t = 0.60$ (Collapsed)

*Figure 13.* Spectral Gap Collapse. Comparison of Hessian spectra at noise scales where separation holds (Left) versus where it fails (Right). In the collapsed states, the normal eigenvalues (right cluster) merge with the tangent eigenvalues, rendering our LHSD ineffective.

## F. Score Network Specifications

To approximate the score function $\nabla_{\mathbf{x}} \log p_t(\mathbf{x})$, we parameterized the score network $\mathbf{s}_\theta(\mathbf{x}, t)$ using two distinct architectures depending on the dimensionality of the target data distribution.

### F.1. MLP-UNet

For datasets with ambient dimensionality $D = 3$, such as the Funnel and Moon manifolds, we used a fully connected network with a UNet-like architecture. We used the implementation of MLP-UNet provided by Kamkari et al. (2024b). This architecture features skip connections between corresponding encoder and decoder layers to preserve high-frequency details. The network configuration and training hyperparameters are detailed in Table 5.

*Table 5.* Hyperparameters for the MLP-UNet architecture used on low-dimensional datasets ($D = 3$).

| Parameter | Value |
|---|---|
| Input Dimension | 3 |
| Hidden Layer Sizes | (1024, 1024, 512, 512, 256, 128) |
| Time Embedding Dimension | 128 |
| Optimizer | Adam |
| Learning Rate | $1 \times 10^{-4}$ |
| Training Epochs | 100 |

### F.2. Conv2D-UNet

For all datasets with ambient dimensions in the range $100 \leq D \leq 3072$, we employed a 2D Convolutional UNet architecture, utilizing the implementation provided by the `diffusers` library.[6] This category includes the IDR-transformed benchmarks, the high-dimensional Manifold Mixtures, and real-world image datasets like CIFAR-10. Data vectors were reshaped into spatial tensors $(C, H, W)$ to be processed by the network. For instance, IDR-FMNIST data ($D = 784$) was reshaped to $1 \times 28 \times 28$ and padded to $32 \times 32$, while CIFAR-10 ($D = 3072$) used its native $3 \times 32 \times 32$ format. The architecture incorporates attention mechanisms at the lowest resolutions to capture global dependencies. The specific configuration is provided in Table 6.

*Table 6.* Hyperparameters for the Conv2D UNet architecture used on high-dimensional datasets ($100 \leq D \leq 3072$).

| Parameter | Value |
|---|---|
| Sample Size | Adapted to dataset (e.g., $32 \times 32$) |
| Input/Output Channels | 1 or 3 |
| Layers Per Block | 2 |
| Block Output Channels | (128, 256, 256) |
| Downsampling Blocks | `Down, Down, AttnDown` |
| Upsampling Blocks | `AttnUp, Up, Up` |
| Time Embedding Factor | 1000 |
| Optimizer | Adam |
| Learning Rate | $1 \times 10^{-4}$ |
| Training Epochs | 500 |

## G. Dataset Specifications

We employ a diverse set of benchmarks to evaluate LID estimators, ranging from high-dimensional mixtures to specific low-dimensional geometric shapes. Additionally, we assess robustness to domain shifts by embedding these manifolds into image space using the Intrinsic Dimension Recovery (IDR) method.

---

[6]huggingface.co/docs/diffusers/index

## G.1. High-Dimensional Manifold Mixtures

We utilize the high-dimensional manifold mixture benchmark framework introduced by Kamkari et al. (2024b), generating the datasets according to their protocol and official code implementation. This dataset defines a distribution as a mixture of $K$ distinct manifolds (modes) embedded within a $D$-dimensional ambient space. Each mode $k$ is associated with an intrinsic dimension $d_k$ and a centroid $\mathbf{c}_k$. We sample latent vectors $\mathbf{z} \sim \mathcal{U}(-1,1)^{d_k}$ from a uniform distribution, which are then transformed into the ambient space via an affine projection $A$ and a subsequent diffeomorphism $f$:

$$\mathbf{x} = f(A(\mathbf{z})) + \mathbf{c}_k \tag{24}$$

The centroids $\mathbf{c}_k$ are positioned such that the pairwise Euclidean distance between modes is at least 20 to maintain separation.

**Linear Projection ($\mathcal{L}$).** The ambient diffeomorphism $f$ is set to the identity map, and the projection $A$ is defined as a random rotation matrix with orthonormal columns. In our results, $\mathcal{L}^{\{d_k\}}$ denotes a mixture generated using this affine projection with component intrinsic dimensions $\{d_k\}$ (e.g., $\mathcal{L}^{10+30+90}$).

**Nonlinear Projection ($\mathcal{F}$)[7].** This setting introduces geometric complexity through non-linearity. The function $f$ is defined as a composition of $T = 5$ sinusoidal transformations. The transformation operates on the ambient vector $\mathbf{h} \in \mathbb{R}^D$ and is defined iteratively as:

$$\mathbf{h}_{t+1} = \mathbf{h}_t + \frac{0.5}{\omega} \sin(\omega \mathbf{h}_t) \tag{25}$$

where $\mathbf{h}_0 = A(\mathbf{z})$ and the frequency $\omega$ controls the complexity of the manifold's curvature. In our results, $\mathcal{F}^{\{d_k\}}$ denotes a mixture generated using this nonlinear sinusoidal projection.

In Table 1, we denote these linear and nonlinear variations as $\mathcal{L}$ and $\mathcal{F}$, respectively.

## G.2. 3D Geometric Manifolds

We reproduce two synthetic benchmarks, Moon and Funnel, originally introduced by Tempczyk et al. (2026). These datasets are embedded in $\mathbb{R}^3$ and challenge estimators with specific geometric features found in real-world data, such as variable dimensionality and boundaries.

**Moon.** The Moon dataset characterizes a manifold with non-zero but small thickness in specific dimensions. Geometrically, this dataset represents a non-convex, connected volume that is moon-shaped in the first two dimensions and extends as a uniform interval in the third dimension, with a height profile dependent on the polar angle. The generation process begins by sampling uniformly from a 2D crescent region $M \subset \mathbb{R}^2$. A point $(x_1, x_2)$ belongs to $M$ if it lies within an outer circle of radius $r$ and outside an inner circle of radius $r_{\text{inner}}$ that is shifted by $\delta_{\text{shift}}$ along the negative $x_2$ axis:

$$M = \left\{ (x_1, x_2) \mid x_1^2 + x_2^2 < r^2 \quad \wedge \quad x_1^2 + (x_2 + \delta_{\text{shift}})^2 > r_{\text{inner}}^2 \right\} \tag{26}$$

The third coordinate, $x_3$, is sampled uniformly from a vertical interval $[-\tau(\phi), \tau(\phi)]$, where the manifold thickness $\tau(\phi)$ is a function of the polar angle $\phi = \text{atan2}(x_2, x_1)$. The thickness profile is defined as $\tau(\phi) = r \cdot [0.001 + 0.1(1 - \sin(\phi))]$ This results in a 3D volume defined by $\mathbf{x} = s \cdot [x_1, x_2, x_3]^\top + \mathbf{c} + \eta$. The dataset assigns discrete integer LIDs based on the proximity of a point to the manifold's boundaries. Points are classified into three categories:

- **Interior ($d_{GT} = 3$):** Points strictly inside the volume, far from all boundaries.

- **Surface ($d_{GT} = 2$):** Points within a tolerance $\epsilon$ of exactly one type of boundary (either the cylindrical walls or the top/bottom caps).

- **Edge ($d_{GT} = 1$):** Points located at the intersection of a cylindrical wall and a top/bottom surface.

**Funnel.** The Funnel dataset simulates a scenario where the local intrinsic dimension (LID) varies continuously across the manifold structure. It constructs a 2D manifold embedded in $\mathbb{R}^3$ that features a smooth geometric transition from a 1D-like "stick" region to a 3D-like "skirt" region. The manifold is generated as a surface of revolution defined by a latent longitudinal

---

[7]We note a discrepancy between the methodology described by Kamkari et al. (2024b) and their official code release: while their paper describes using Neural Spline Flows to generate nonlinear manifolds, their implementation utilizes a composition of sinusoidal diffeomorphisms.

parameter $t \sim \mathcal{U}(t_{\min}, t_{\max})$ and an angular parameter $\theta \sim \mathcal{U}(0, 2\pi)$. The radius of the funnel decays exponentially along the principal axis according to $r(t) = r_0 \exp(-t)$. The base coordinates $\mathbf{x}_{\text{base}} \in \mathbb{R}^3$ are given by:

$$\mathbf{x}_{\text{base}} = \begin{bmatrix} t - t_{\text{shift}} \\ r(t) \sin(\theta) \\ r(t) \cos(\theta) \end{bmatrix} \tag{27}$$

To introduce stochasticity and volume, the final samples $\mathbf{x}$ are subject to a global scaling factor $s$, translation by $\mathbf{c}$, and additive Gaussian noise $\eta \sim \mathcal{N}(0, \sigma_{\text{noise}}^2 \mathbf{I})$.

Ground truth LID assignments, denoted $d_{GT}(\mathbf{x})$, are computed based on the local radius $r(t)$. We define two thresholds, $r_{\text{stick}}$ and $r_{\text{skirt}}$. Points with a sufficiently small radius ($r \leq r_{\text{stick}}$) effectively form a 1D line and are assigned $d_{GT} = 1$. Conversely, points with a large radius ($r \geq r_{\text{skirt}}$) are treated as a 3D volume with $d_{GT} = 3$. For intermediate radii, the dimensionality transitions continuously between 1 and 3 using a cubic Hermite interpolation function.

### G.3. Intrinsic Dimension Recovery (IDR) Method

Intrinsic Dimension Recovery (IDR) is a transformation method introduced by Tempczyk et al. (2026) designed to embed arbitrary low-dimensional manifolds into high-dimensional, realistic data domains. We apply IDR to the Moon, Funnel, and the affine manifold mixture $\mathcal{L}^{1+2+3}$ to generate high-dimensional image benchmarks. The embedding process maps coordinates from the base manifold $\mathbf{x} \in \mathbb{R}^d$ into the pixel space of Fashion-MNIST (FMNIST) images $\mathbf{I} \in \mathbb{R}^{784}$ via a non-linear projection. The procedure consists of three steps:

1. **Basis Construction:** We compute the Principal Component Analysis (PCA) of a single FMNIST class (Class 7: Sneakers) to extract a mean image $\boldsymbol{\mu} \in \mathbb{R}^{784}$ and a basis matrix of principal components $\mathbf{U} \in \mathbb{R}^{784 \times K}$. Using a single class ensures the generated images possess a coherent visual structure.

2. **Random Fourier Feature Mapping:** To ensure the embedding is non-linear and smooth, the low-dimensional input $\mathbf{x}$ is mapped to a higher-dimensional feature vector $\phi(\mathbf{x}) \in \mathbb{R}^K$. This is achieved using Random Fourier Features (RFF) combined with bias terms:

$$\phi(\mathbf{x}) = \left[ \sin(\mathbf{W}\mathbf{x} + \mathbf{b}), \cos(\mathbf{W}\mathbf{x} + \mathbf{b}), 1, \|\mathbf{x}\|^2 \right] \tag{28}$$

   where $\mathbf{W} \in \mathbb{R}^{K' \times d}$ and $\mathbf{b} \in \mathbb{R}^{K'}$ are fixed weights drawn from a normal and uniform distribution, respectively. This mapping effectively "wraps" the low-dimensional manifold into the latent space of the image basis.

3. **Image Projection:** The feature vector is projected back into the image space using the pre-computed PCA basis:

$$\mathbf{I} = \text{clamp}\left( \boldsymbol{\mu} + \phi(\mathbf{x})\mathbf{U}^\top, 0, 1 \right) \tag{29}$$

Since the intrinsic dimension is invariant under diffeomorphisms, the topological properties of the data are preserved. Therefore, we define the ground truth LID of the IDR-transformed samples to be identical to the LID of their corresponding generating points in the base manifold.

## H. Impact of Lanczos Steps ($m$)

We investigate the optimal choice for the number of Lanczos steps, $m$. Referring to Table 7, we observe that setting $m = 5$ yields the lowest average MAE of 6.60, significantly outperforming the $m = 2$ setting overall, although $m = 2$ can be slightly better in some simple linear-manifold cases where the tangent/normal spectrum is already clean and a shallow Lanczos approximation captures the main structure needed for LID estimation.

Increasing $m$ further to 10 or beyond provides little additional gain and can even slightly degrade performance, suggesting that the approximation has largely saturated. A plausible reason is finite-precision loss of orthogonality in the Lanczos basis (Chen et al., 2021; Meurant & Strakoš, 2006): while the basis vectors should remain orthonormal in exact arithmetic, this property can gradually deteriorate numerically as $m$ increases, causing later vectors to partially re-enter previously discovered Ritz directions. This may produce duplicated or spurious Ritz values and make $e_1^\top f(T_m) e_1$ less stable, thereby degrading downstream MAE. By contrast, for small $m$, the method may be less sensitive to such numerical effects.

Next, examining the detailed trajectories in Fig. 14 (a–d), we confirm that this superiority holds consistently across the entire noise scale range $0 < t < 0.2$. Conversely, as shown in Fig. 14 (e), the inference time increases linearly with $m$. Based on these observations, we conclude that $m = 5$ offers the optimal balance between accuracy and computational efficiency.

*Table 7.* Impact of Lanczos steps $m$ on estimation accuracy (MAE). While $m = 2$ provides a fast approximation, increasing $m$ to 5 significantly reduces error. Further increasing $m$ beyond 5 yields diminishing returns.

| Method | $D = 100$ | | $D = 1024$ | | | $D = 3072$ | | | | Avg. |
|---|---|---|---|---|---|---|---|---|---|---|
| | $\mathcal{L}^{10+30+90}$ | $\mathcal{F}^{10+25+50}$ | $\mathcal{L}^{900}$ | $\mathcal{L}^{10+80+200}$ | $\mathcal{F}^{10+80+200}$ | $\mathcal{L}^{900}$ | $\mathcal{F}^{900}$ | $\mathcal{L}^{10+80+200}$ | $\mathcal{F}^{10+80+200}$ | |
| LHSD ($m = 2$) | **1.59** | 7.66 | **4.77** | 8.97 | 21.30 | 35.00 | 39.96 | 37.64 | 29.70 | 20.7 |
| LHSD ($m = 5$) | 1.64 | 2.16 | 5.03 | 3.47 | 6.90 | **11.53** | **18.79** | **4.70** | **5.19** | **6.6** |
| LHSD ($m = 10$) | 1.69 | 2.03 | 4.90 | 3.61 | **4.54** | 11.64 | 36.36 | 5.39 | 7.02 | 8.6 |
| LHSD ($m = 20$) | 1.64 | **2.02** | 5.05 | **3.33** | 4.85 | 11.97 | 40.42 | 5.48 | 6.68 | 9.0 |

### H.1. Justification for the Saturation at $m = 5$

To understand why the accuracy improves significantly up to $m = 5$ and saturates thereafter, we analyze the approximation capacity of SLQ by relating the number of Lanczos iterations to the effective polynomial expressivity of the resulting quadrature. An $m$-step Lanczos procedure (see App. B for details) yields a tridiagonal matrix whose associated Gaussian quadrature exactly matches moments up to degree $2m - 1$. This implies that quadratic forms of any polynomial function of degree at most $k = 2m - 1$ are evaluated exactly.

We consider the local Taylor expansion of the Hill-type filter with steepness $p = 4$ (Eq. 9) around $\lambda = 0$:

$$f(\lambda) = \frac{1}{1 + (\frac{\lambda}{\kappa})^4} = 1 - \left(\frac{\lambda}{\kappa}\right)^4 + \left(\frac{\lambda}{\kappa}\right)^8 - \mathcal{O}(\lambda^{12}). \tag{30}$$

This expansion indicates that, within the pass-band region ($\lambda \ll \kappa$), the leading deviations from unity are governed by the fourth- and eighth-order terms.

- *Case $m = 2$ (Max Degree $k = 3$):* With $m = 2$, the effective polynomial expressivity is limited to cubic order. Consequently, the approximation implies $f(\lambda) \approx 1$ locally, failing to represent higher-order curvature effects such as the $\lambda^4$ term in Eq. (30). When a non-negligible portion of the spectral mass lies near the transition region of the filter, this limited expressivity hinders the quadrature's ability to accurately capture the initial roll-off behavior, leading to the higher approximation errors observed in our experiments.

- *Case $m = 5$ (Max Degree $k = 9$):* Increasing to $m = 5$ extends the effective expressivity to ninth order. This allows the approximation to capture both the leading $\lambda^4$ term and the secondary $\lambda^8$ correction, providing sufficient flexibility to model the filter's curvature near its transition. Empirically, this increased expressivity explains the substantial accuracy gains, as the quadrature can correctly weigh eigenvalues populating the vicinity of the cutoff.

However, the limitation of $m = 2$ does not imply that the estimator reduces to a trivial value. Although the local Taylor expansion suggests $f(\lambda) \approx 1$ pointwise near the origin, SLQ approximates the *spectral integral* $\int f(\lambda) \, d\mu(\lambda)$ rather than

the function $f(\lambda)$ itself. When the Hessian spectrum is strongly polarized into near-zero (tangent) and large (normal) eigenvalues, as assumed in Property 1, even a low-rank quadrature can yield a reasonable estimate by effectively aggregating the spectral mass of these two distinct regimes. The higher accuracy at $m = 5$ arises from the ability to refine this aggregation by resolving the spectral mass located within the filter's transition region.

In summary, since the dominant spectral features (the sharp transition defined by $p = 4$) are structurally captured by a 9th-degree polynomial ($m = 5$), further increasing $m$ provides diminishing returns, aligning with the saturation observed in Table 7 and Fig. 14.

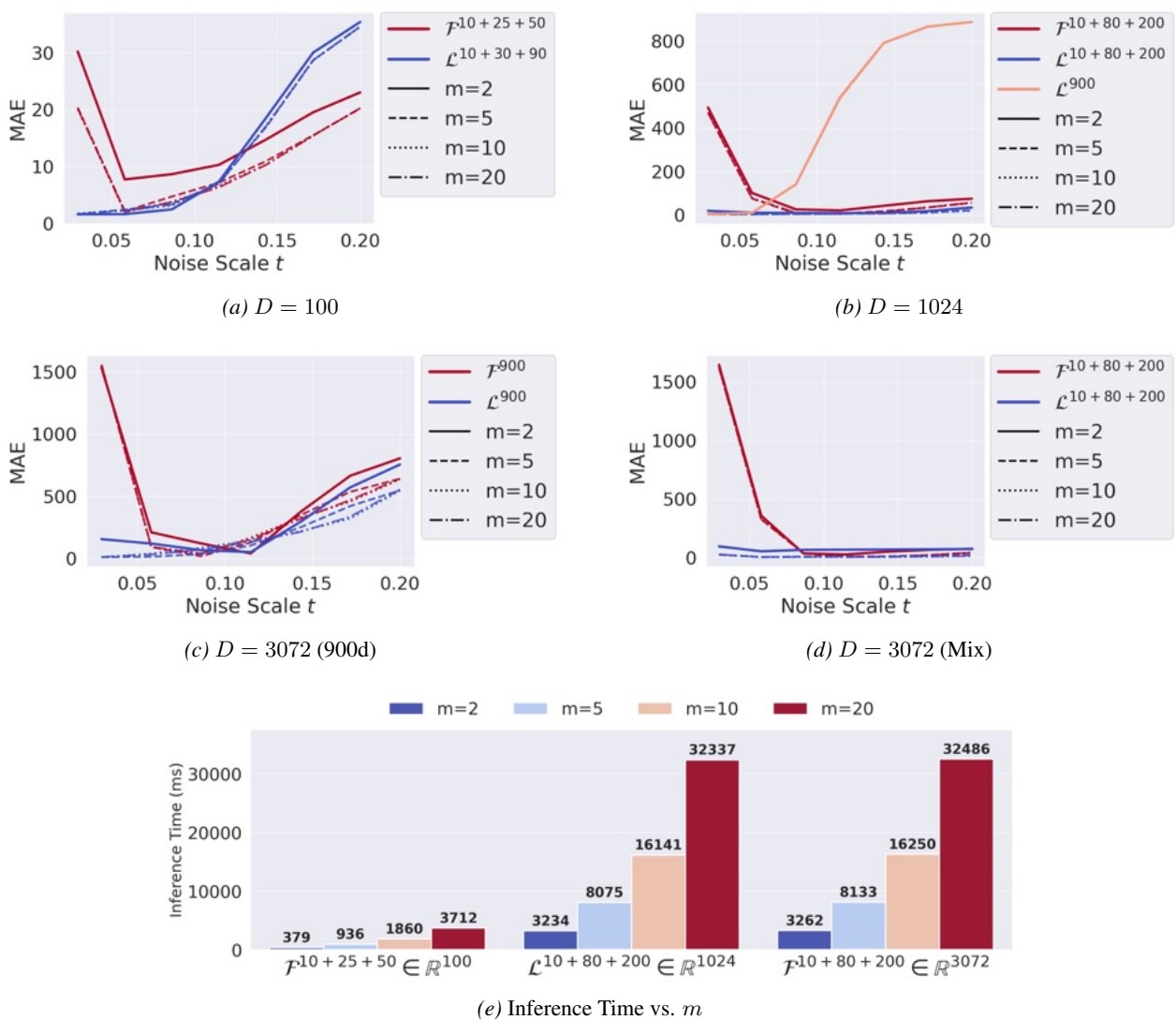

*Figure 14.* Sensitivity analysis of LHSD with respect to Lanczos steps $m$. Top panels (a-d) show the MAE across noise scales $t$ for varying $m$. The bottom panel (e) compares the inference time. Increasing $m$ improves accuracy up to a saturation point around $m = 5$, beyond which computational cost increases without significant gain.

# I. Distribution of Estimated Local Intrinsic Dimensions

We show the distribution of LID estimated by LHSD in the high-dimensional manifold experiments described in Sec. 5 (Experiment 1). In these evaluations, the number of Lanczos steps was set to $m = 5$. As shown in Fig. 15, LHSD captures the intrinsic structure of the data manifolds across all settings. For instance, in the multi-manifold case $\mathcal{L}^{10+80+200}$, the estimated LIDs form distinct clusters around the ground-truth dimensions of 10, 80, and 200. Similarly, for the single-manifold case $\mathcal{L}^{900}$, the distribution is tightly concentrated around 900. Overall, the estimated distributions consistently align with the ground-truth dimensions across the datasets.

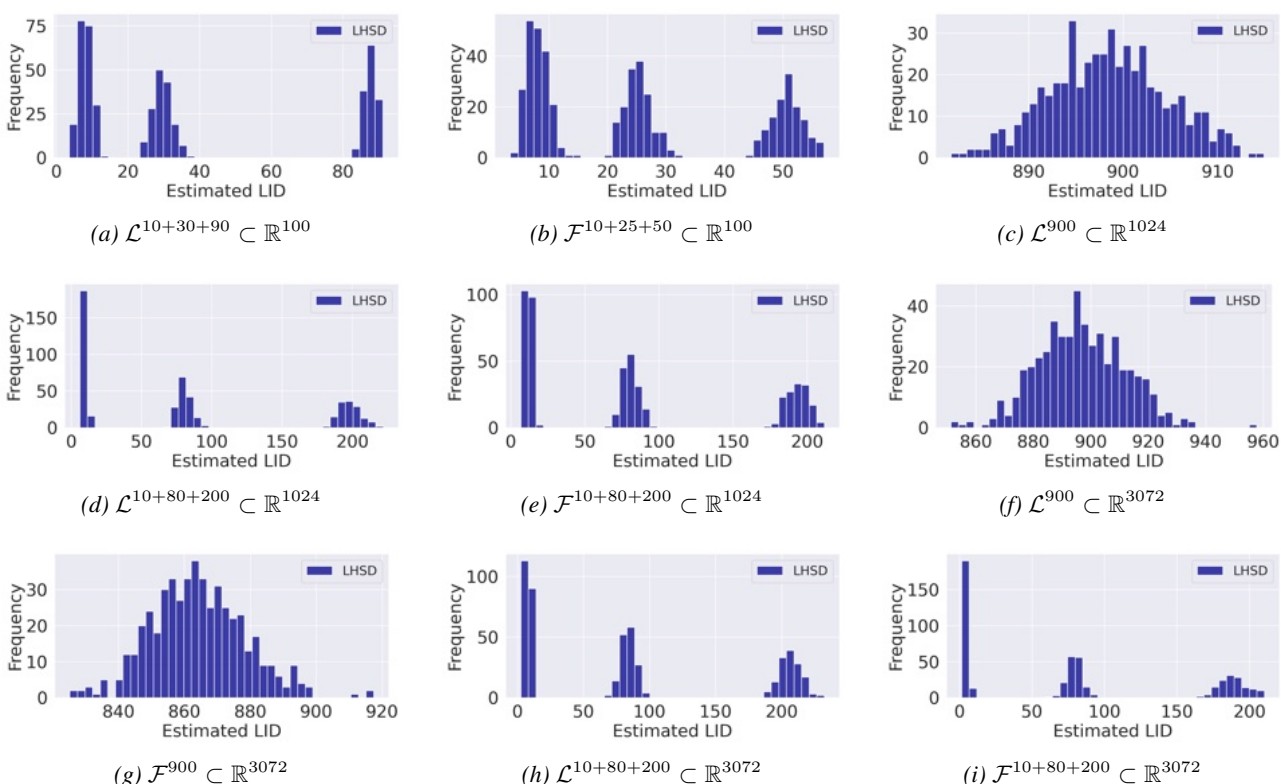

*(a) $\mathcal{L}^{10+30+90} \subset \mathbb{R}^{100}$*  *(b) $\mathcal{F}^{10+25+50} \subset \mathbb{R}^{100}$*  *(c) $\mathcal{L}^{900} \subset \mathbb{R}^{1024}$*

*(d) $\mathcal{L}^{10+80+200} \subset \mathbb{R}^{1024}$*  *(e) $\mathcal{F}^{10+80+200} \subset \mathbb{R}^{1024}$*  *(f) $\mathcal{L}^{900} \subset \mathbb{R}^{3072}$*

*(g) $\mathcal{F}^{900} \subset \mathbb{R}^{3072}$*  *(h) $\mathcal{L}^{10+80+200} \subset \mathbb{R}^{3072}$*  *(i) $\mathcal{F}^{10+80+200} \subset \mathbb{R}^{3072}$*

*Figure 15.* Histograms of LID estimated by LHSD. The peaks of the distributions closely align with the ground-truth dimensions of the underlying submanifolds (e.g., peaks at 10, 80, and 200 for $\mathcal{L}^{10+80+200}$), demonstrating the accuracy of the estimation.

# J. Details of Runtime Measurement Experiments

**Stochastic FLIPD (FLIPD-Hutch).** The official implementation of FLIPD (Kamkari et al., 2024b) is a deterministic algorithm that exactly computes the diagonal elements of the Hessian. To compare computational efficiency fairly with LHSD, we implemented a stochastic variant of FLIPD (referred to as *FLIPD-Hutch* in this paper) that utilizes Hutchinson estimation. Specifically, we replaced the process where the original FLIPD performs $D$ backpropagations with Hessian-Vector Products (HVP) using $K = 8$ random Rademacher vectors. Consequently, the algorithmic computational complexity becomes of the same order as that of LHSD.

**Measurement Protocol.**

Inference times were measured on a single NVIDIA A100 GPU using 500 samples from the Moon, Funnel, and $\mathcal{L}^{1+2+3}$ datasets of the IDR benchmark. To accurately capture asynchronous GPU execution, we used `torch.cuda.Event` and applied `torch.cuda.synchronize()` before and after the measurement interval to ensure completion. The reported values are the average execution time (in milliseconds) of 10 consecutive runs following two warm-up runs.

**Note on Runtime Discrepancy.**

As shown in Fig. 5 left, LHSD is approximately $2.5\times$ faster than FLIPD-Hutch. Although both methods employ a comparable

number of HVPs, we attribute this speed difference primarily to the automatic differentiation (AD) modes adopted by each method. The implementation of FLIPD relies on Forward-mode AD (Jacobian-Vector Product, JVP). However, under current PyTorch specifications, acceleration kernels such as Flash Attention must be disabled to maintain numerical consistency when using Forward-mode AD, which becomes a source of latency. In contrast, LHSD utilizes standard Reverse-mode AD (Vector-Jacobian Product, VJP/Backpropagation), allowing it to fully benefit from these hardware accelerations. It is worth noting that even if future optimizations resolve this speed discrepancy, the superiority of LHSD remains unshaken because, as demonstrated in Fig. 5 right, the estimation accuracy of FLIPD-Hutch degrades significantly due to variance explosion.

## K. Sensitivity Analysis regarding Noise Scale $t$

We visually demonstrate the sensitivity of the estimation accuracy to the noise scale (diffusion time) parameter $t$ for LHSD (Ours), FLIPD, and NB, which are methods dependent on the diffusion process time. For the experiments, we used the three manifolds from the IDR benchmark (Moon, Funnel, and $\mathcal{L}^{1+2+3}$). Estimations were performed at 16 distinct points within the range $t \in [10^{-3}, 10^{-1}]$. The estimation was conducted in the 784-dimensional space (ambient space of FMNIST) where the data was embedded according to the IDR protocol; subsequently, the results were mapped back to the original 3-dimensional space for visualization. We used 500 samples for each dataset. The color bars adjacent to the plots indicate the dynamic range (minimum to maximum) of the estimated values at each time step.

The results are presented in Figs. 16 to 24. LHSD consistently maintains estimates near the true dimension across almost the entire range of $t$, demonstrating exceptional robustness to parameter selection. In contrast, FLIPD and NB exhibit high sensitivity to changes in $t$, with dramatic fluctuations observed in the estimated values. Notably, NB tends to yield overestimated values in the small-$t$ regime. This suggests that the method is overly sensitive to minute noise components in the high-dimensional ambient space.

Furthermore, it is noteworthy that FLIPD records negative estimates up to approximately $t < 0.06$. While LID must theoretically satisfy $0 \leq d \leq D$, FLIPD and LIDL, which are based on the rate of density change, incorporate the score divergence $\nabla \cdot \mathbf{s}_\theta$ (equivalent to the Laplacian of the log-density $\Delta \log p$) into their estimators. In high-dimensional and high co-dimension settings, the sharp drop in density along the normal directions, which occupy the vast majority of the dimensions, causes this term to take massive negative values, resulting in the output of negative dimensions. This leads to severe instability. While one may post-process by clamping the output, this does not address the underlying instability: the estimator is dominated by indiscriminate sums over the high-dimensional normal subspace, which can overwhelm the signal and lead to unphysical outputs.

## L. Dataset Construction for Memorization Detection

The specific procedure for constructing the memorization dataset largely follows the protocol of Ross et al. (2025). First, we generated 200,000 images using the DDPM and extracted the top images with high similarity to the training data based on SSCD and the calibrated $\ell_2$ distance (Pizzi et al., 2022; Carlini et al., 2023). We manually inspected these candidates and identified 126 images exhibiting memorization, designating them as the memorization set $\mathcal{S}_{\text{mem}}$.

Next, we constructed the comparative non-memorized set $\mathcal{S}_{\text{non}}$. Visual complexity acts as a confounding factor in LID estimation (i.e., simple images tend to have lower LID). To control for this, we used the mean PNG complexity (measured by the bit length after encoding) of $\mathcal{S}_{\text{mem}}$ as a baseline. We extracted non-memorized images only within a difference of $\pm 1992$ bits from this baseline, thereby adjusting the complexity distributions of $\mathcal{S}_{\text{mem}}$ and $\mathcal{S}_{\text{non}}$ to overlap.

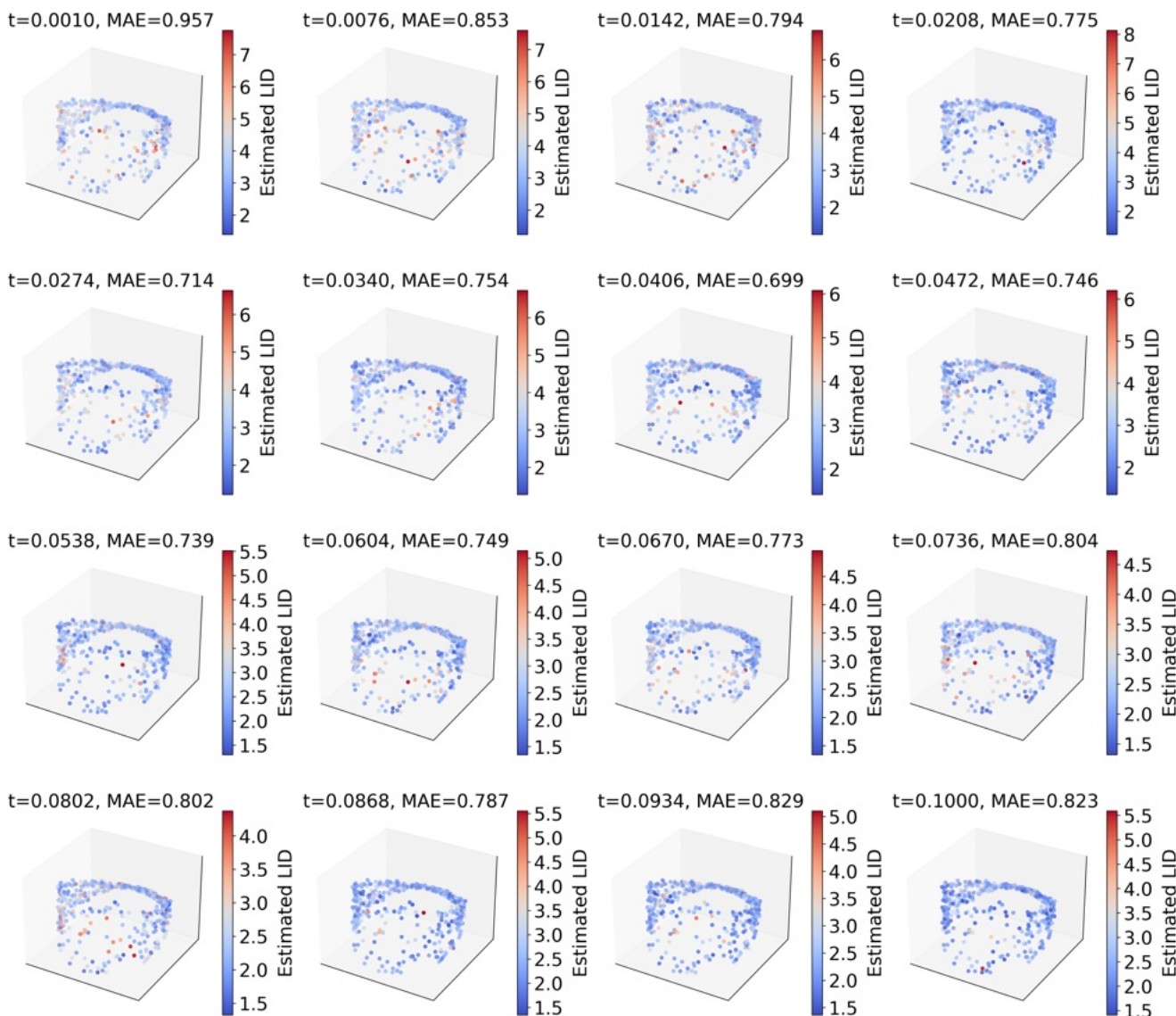

*Figure 16.* Noise scale sensitivity: LHSD (ours) on Moon.

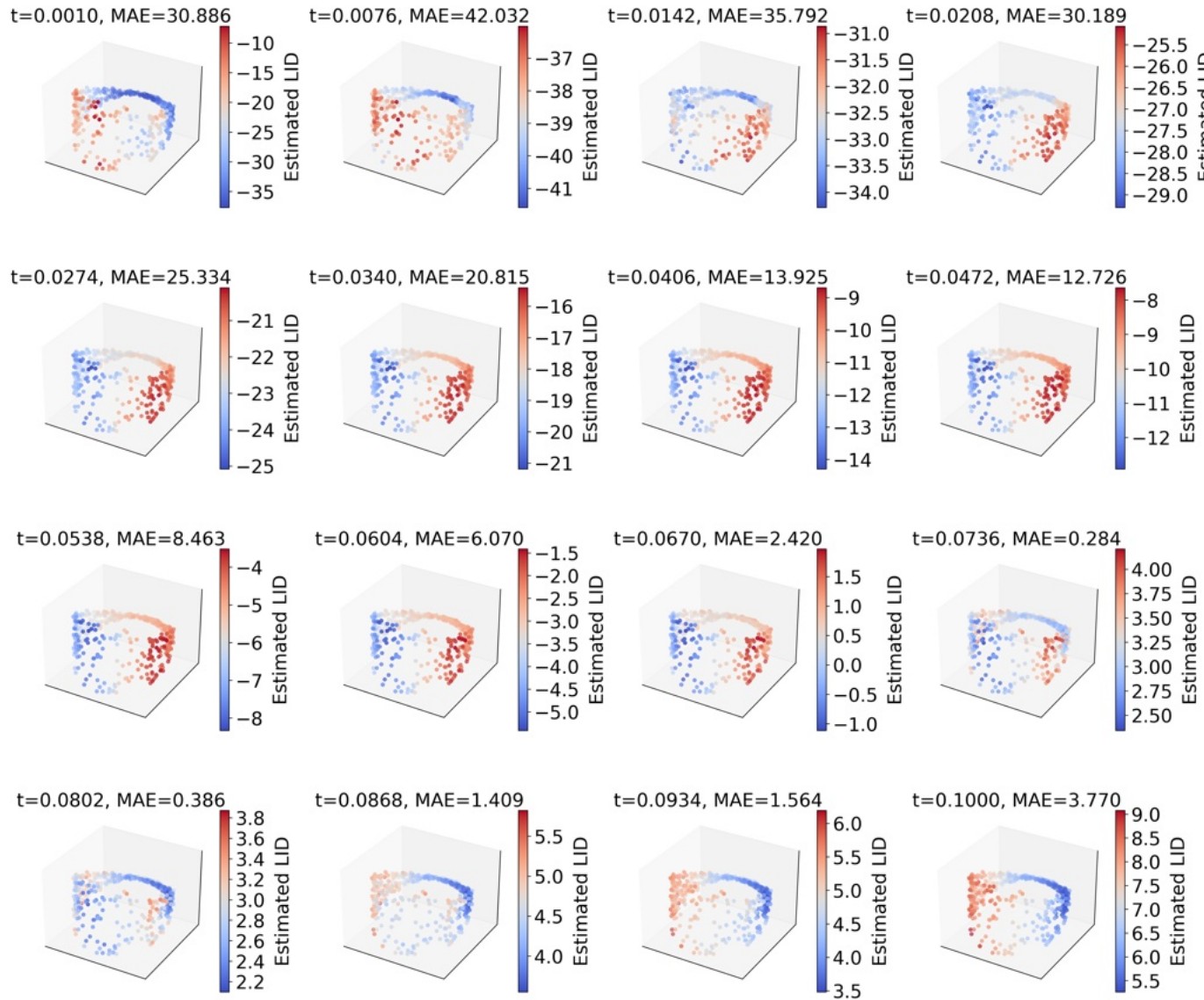

*Figure 17.* Noise scale sensitivity: FLIPD on Moon.

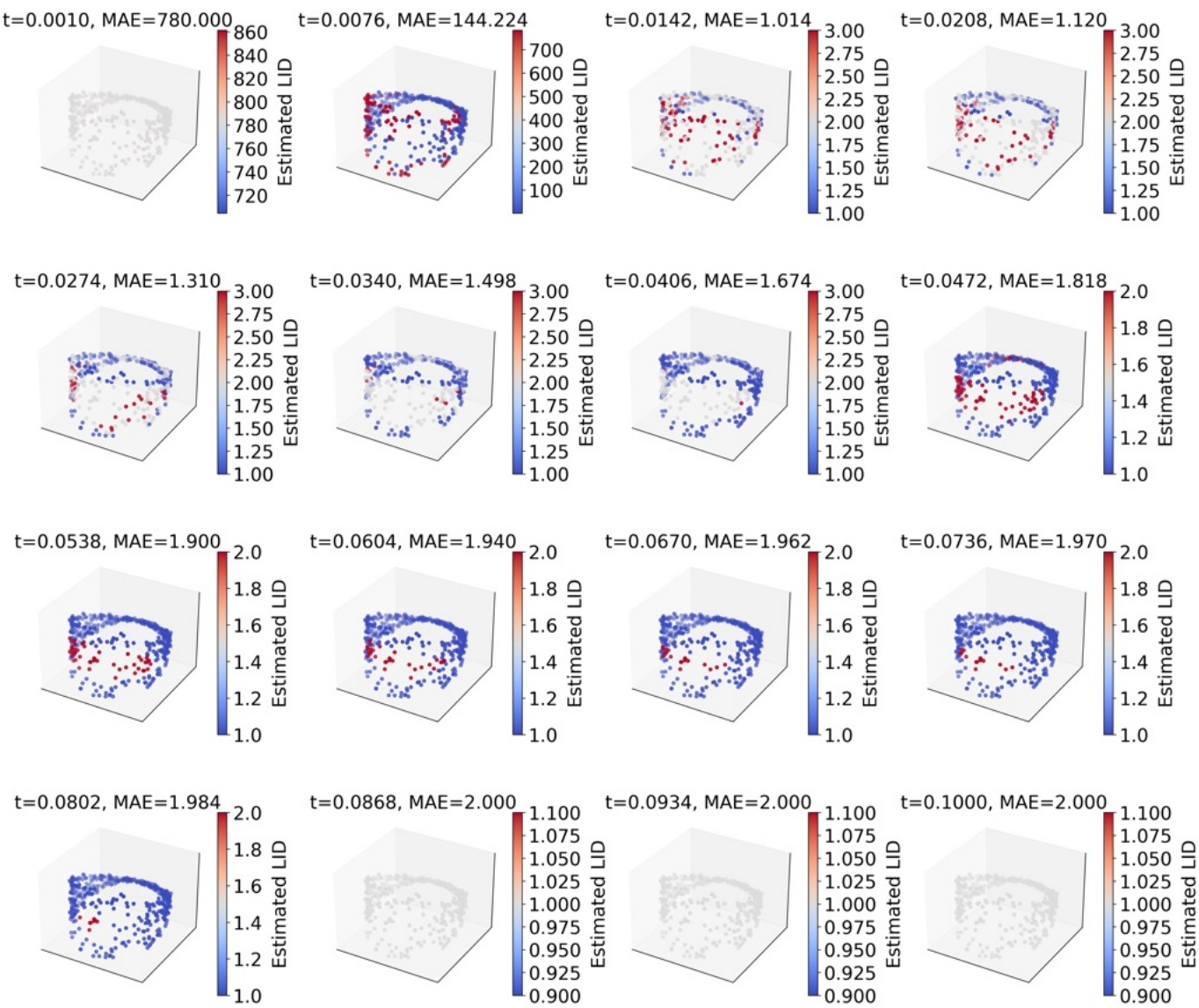

*Figure 18.* Noise scale sensitivity: NB on Moon.

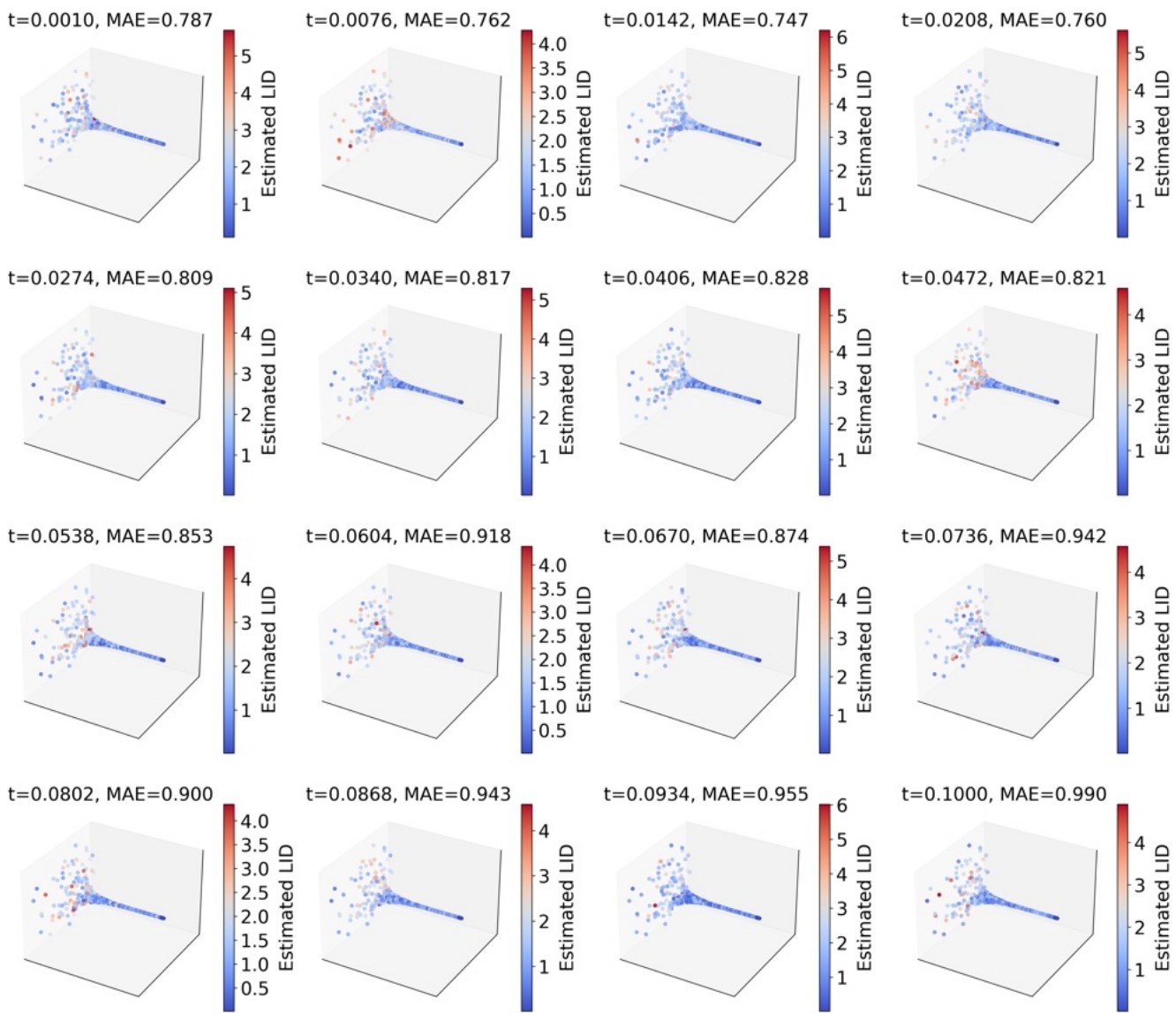

*Figure 19.* Noise scale sensitivity: LHSD (ours) on Funnel.

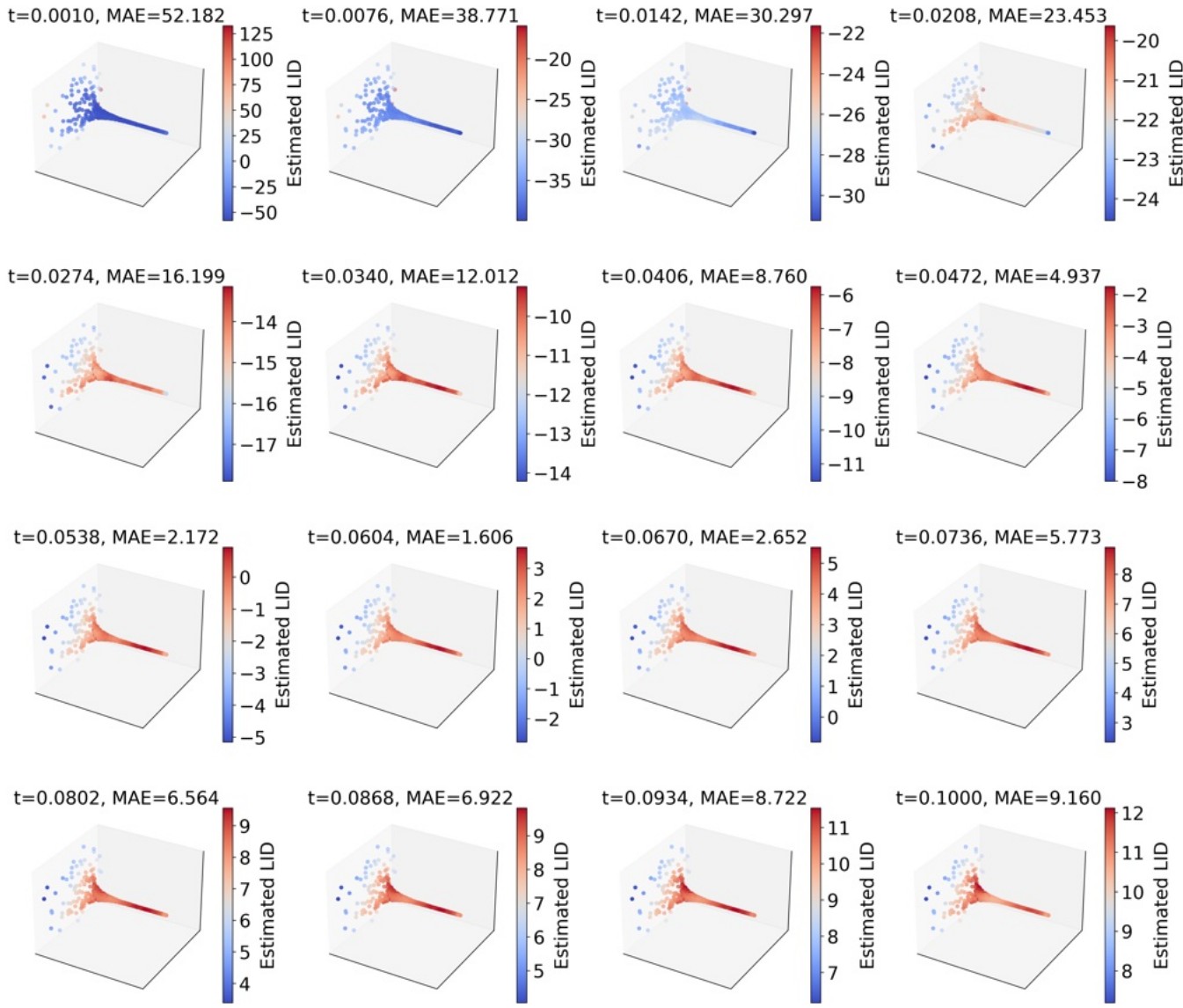

*Figure 20.* Noise scale sensitivity: FLIPD on Funnel.

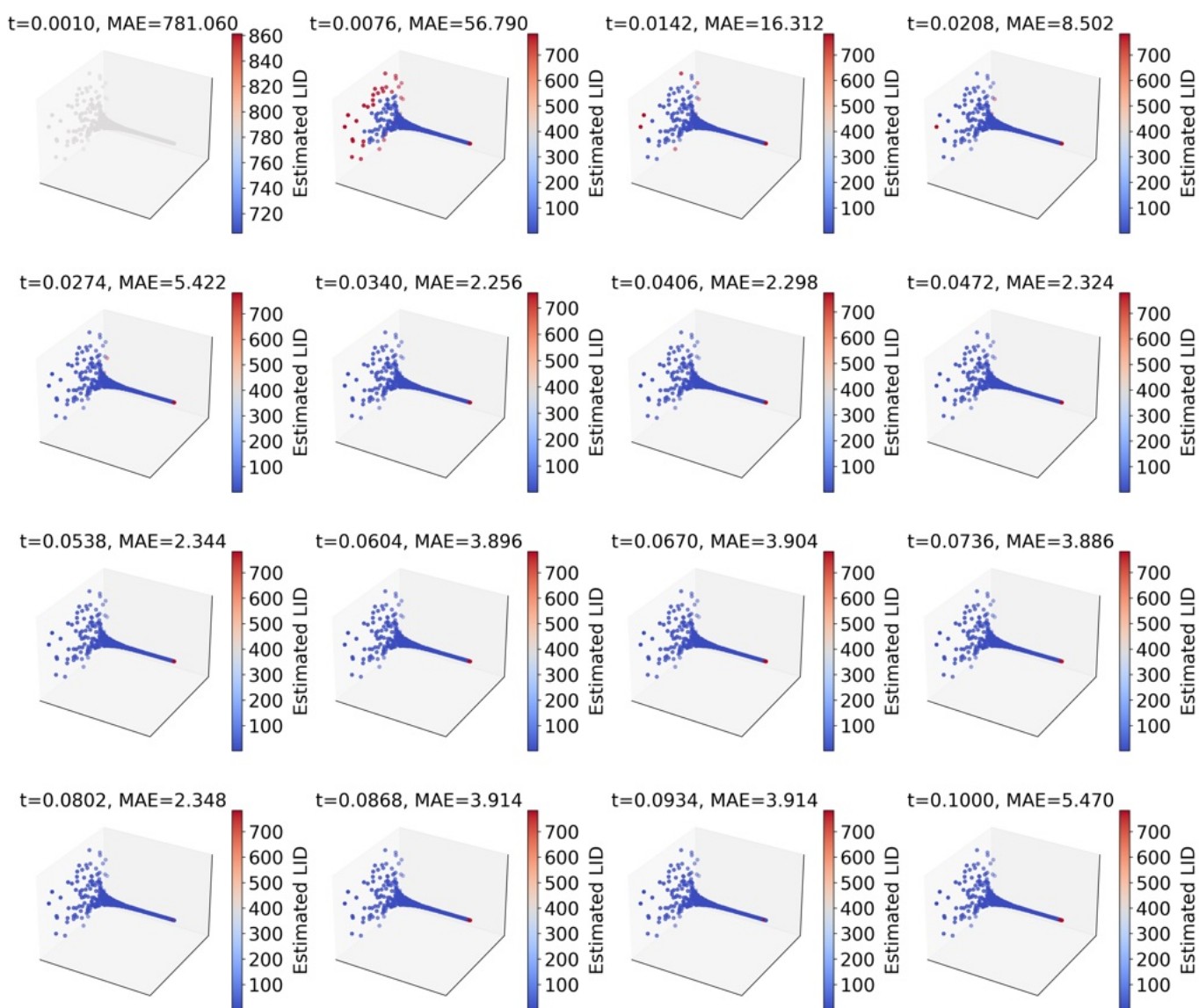

*Figure 21.* Noise scale sensitivity: NB on Funnel.

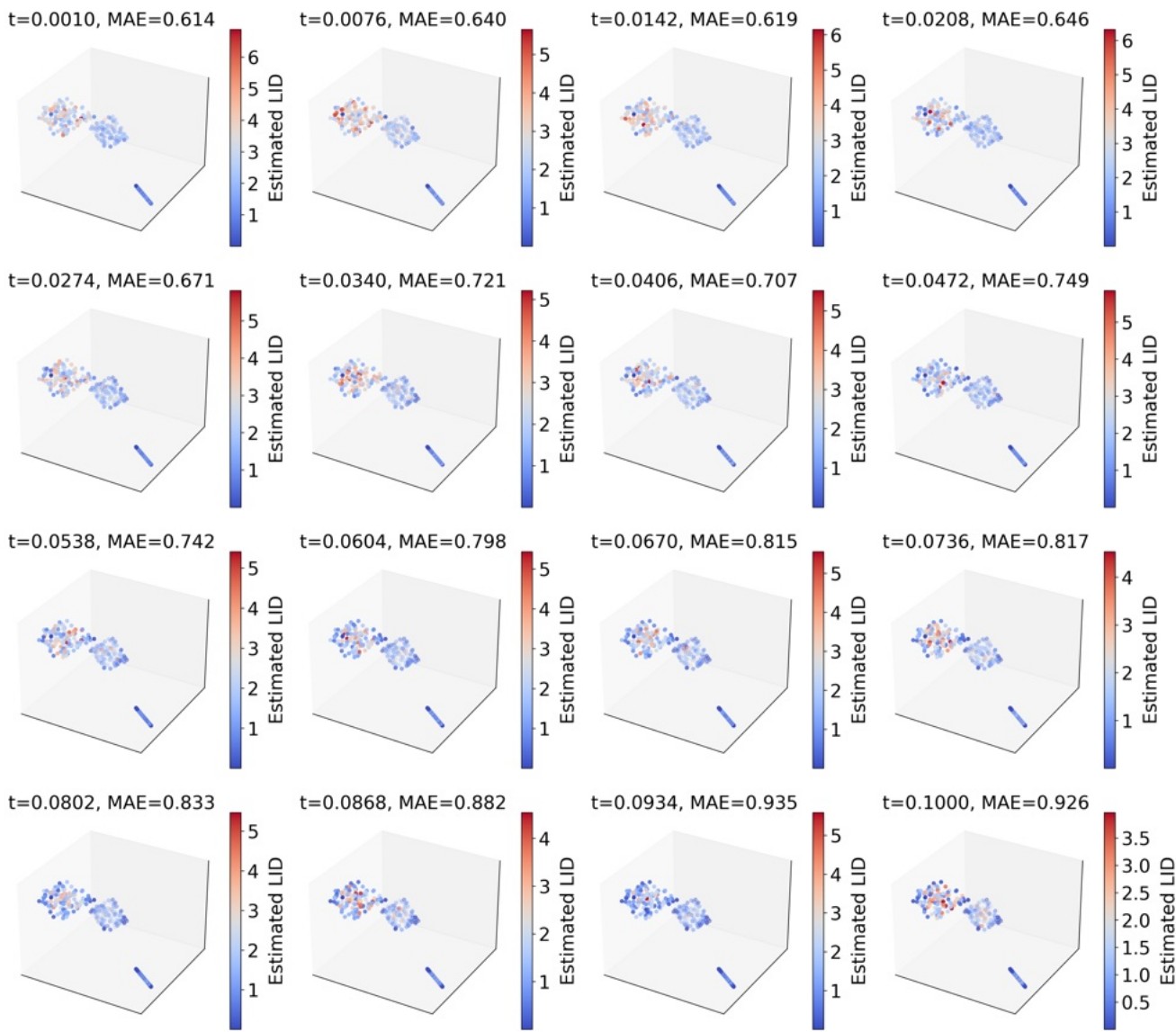

*Figure 22.* Noise scale sensitivity: LHSD (ours) on $\mathcal{L}^{1+2+3}$.

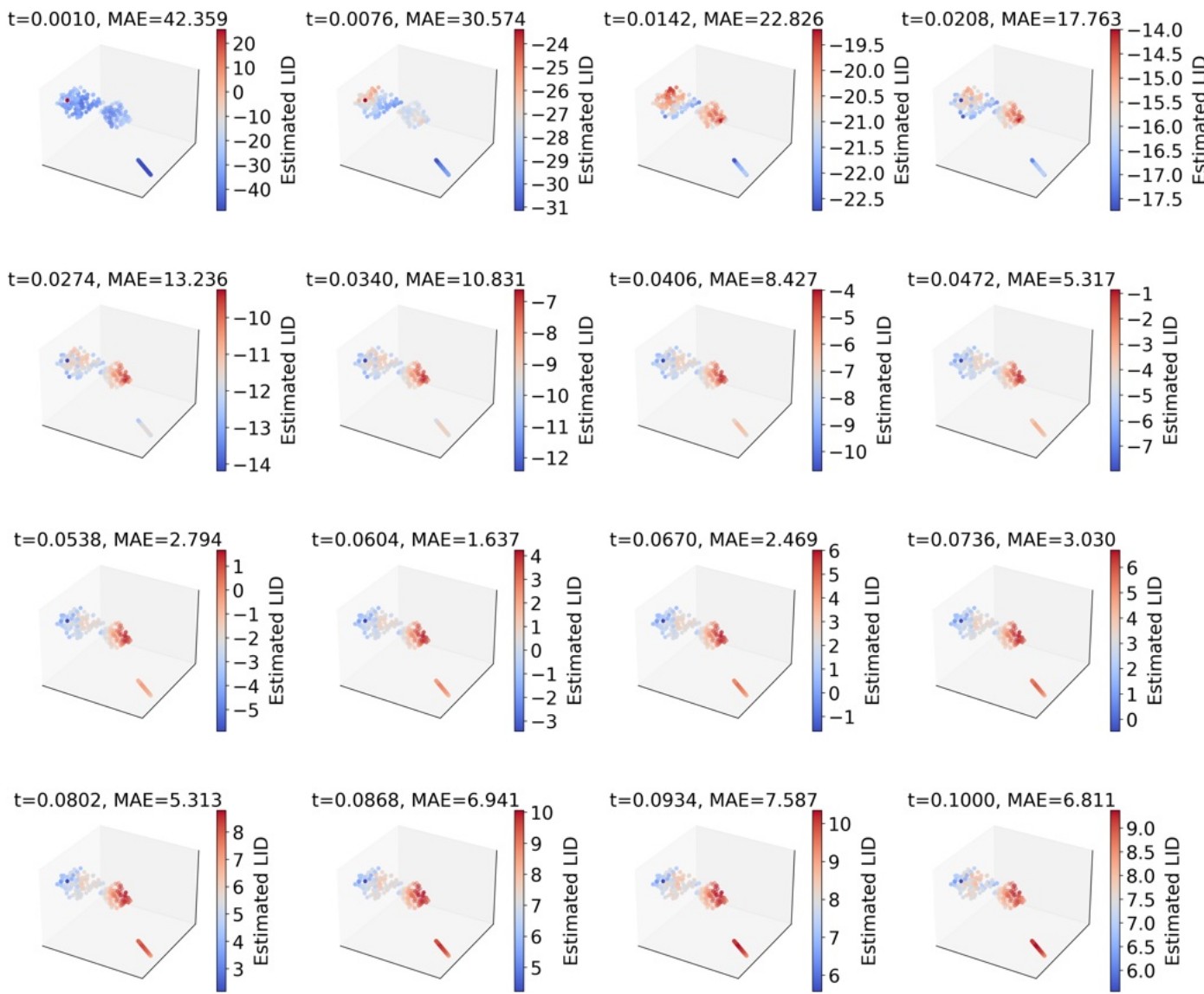

*Figure 23.* Noise scale sensitivity: FLIPD on $\mathcal{L}^{1+2+3}$.

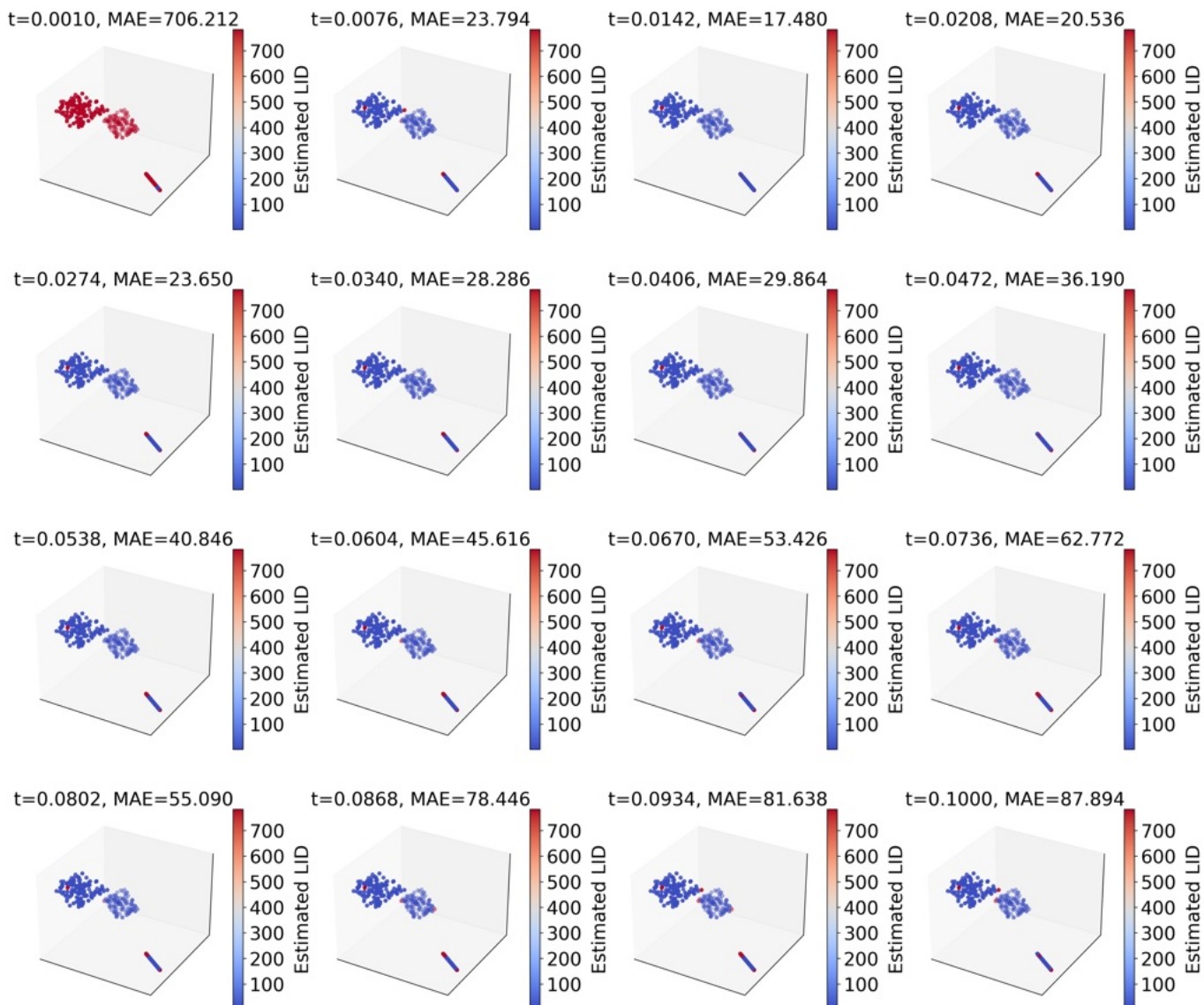

*Figure 24.* Noise scale sensitivity: NB on $\mathcal{L}^{1+2+3}$.

