# OpenReview forum: "Local Hessian Spectral Filtering for Robust Intrinsic Dimension Estimation"
_ICML.cc/2026/Conference — ICML 2026 regular_

### Official Review · Reviewer_FRpa · 2026-03-10

**Soundness:** 3
**Presentation:** 4
**Significance:** 2
**Originality:** 3
**Overall Recommendation:** 5
**Confidence:** 4

**Summary:**

This paper introduces a method for estimating the local intrinsic dimension of data manifolds using the eigenvalues of the hessian of the log-density derived from a trained score model. The principle of the method relies on the fact that the number of zero eigenvalues of this Hessian is equal to the local intrinsic dimension and that if a filtering threshold is chosen judiciously, local intrinsic dimension can be estimated from the eigenvalues of the gradient of a suitably trained score estimator, as common arises in generative models.
The paper explains the details of the method, how it compares to other existing methods, how it can be implemented in practice using Stochastic Lanczos Quadrature, discusses the robustness of the estimator and illustrates its performance in numerical comparisons.

**Compliance With Llm Reviewing Policy:**

Affirmed.

**Final Justification:**

The rebuttal addressed my key questions so I have raised my overall recommendation score from 4 to 5.

**Key Questions For Authors:**

1) In the numerical comparisons against k-nn methods, how many data points in total where used for the k-nn estimators? Is it the N=2000 points mentioned at the top of column 2 on page 6? If so, this seems a very unfair comparison against the generative models being trained on 500k samples! Or did the k-nn methods use all 500k, or perhaps something else? It needs to be completely clear that a fair comparison is being made.

2) It is claimed in several places in the paper that kNN methods are affected by the curse of dimensionality. Please provide references to back this up – and is it something which has been mathematically proven, or observed empirically in the literature, or both? Is the theoretical convergence rate of (any) kNN LID estimator known?

3) It is made clear in the paper that the Lanczos approximation is key to making the approach scalable. The authors point out that the existing Normal Bundle approach involves an SVD and hence has cubic complexity in D. Is it obvious that there is no approximation technique available (such as randomized approximation to SVD) that can obviate this complexity, by analogy to the authors’ use of Lanczos to make their method viable? It’s important to be fair and balanced here.

**Limitations:**

Limitations are explicitly considered in section 6.

**Strengths And Weaknesses:**

As far as I can tell, the paper is mathematically sound. As described in the paper, there have been several recent works on leveraging pre-trained score-models to estimate intrinsic dimension, so this is an active area of research.

I have an important question about the experimental setup (see below) and need to understand their answer to this before having an opinion on the soundness and significance of the numerical results. I have set preliminary scores of 2 for soundness and significance, but my final scores will depend on their answer.

The paper is very well presented and does a great job of efficiently packing a lot of useful and insightful information into the conference paper format. I found almost all the paper very easy to read, which is especially notable considering that some of the mathematical concepts involved are (by the standards of ML/AI conferences) quite sophisticated. Equation (8) and the discussion surrounding it is absolutely central to the work, and I think more clarity is needed here: on first reaching this equation I could see no reason (in the exposition so far) why the inter-change of $tr$ and the filter $f$ is valid (indeed f is not linear in general, as I understand it). A few lines later it is explained that $f(\lambda) $ is approximately 1 or 0 depending on lambda, I was left wondering if this was supposed to justify the final equality in eq (8) – is that what the authors intended? More clarity is needed over what is and is not an approximation.

As such the paper provides a new and seemingly useful LID estimation technique.

---

> ### Author Rebuttal · Authors · 2026-03-28
>
> We thank the reviewer for the careful reading and thoughtful comments. We especially appreciate the positive feedback on the presentation and the helpful identification of two places that need clearer exposition: the derivation around Eq. (8) and the experimental comparison protocol.
>
>
> ---
>
> ## Regarding the derivation of Eq. (8) and the apparent interchange of $\mathrm{tr}$ and the filter $f$:
>
> Eq. (8) does not use an identity of the form $\mathrm{tr}(f(H)) = f(\mathrm{tr}(H))$. Rather, in the theoretical setting, $H(x,t):=-\nabla_x^2 \log p_t(x)$ is symmetric, so with the spectral decomposition $H = U \Lambda U^\top$, the matrix function is defined as $f(H):=U f(\Lambda) U^\top$. Therefore,
> $\mathrm{tr}(f(H)) = \mathrm{tr}(U f(\Lambda) U^\top) = \mathrm{tr}(f(\Lambda)) = \sum_{i=1}^D f(\lambda_i)$,
> which is an exact identity.
>
> The later discussion that $f(\lambda)$ is close to $1$ or $0$ is not meant to justify this equality. Rather, it is meant to justify the approximation $\mathrm{LID}(x) \approx \mathrm{LHSD}(x)$, namely that $f$ acts as a soft indicator of tangent versus normal directions. In practice, two additional approximations enter: (i) replacing the true Hessian by $H_\theta(x,t):=-\nabla_x s_\theta(x,t)$ using a learned score model, and (ii) approximating $\mathrm{tr}(f(H_\theta))$ numerically via SLQ.
>
> We thank the reviewer for pointing out this ambiguity and will clarify in the main text which part is exact and which part is approximate.
>
> ---
>
>
> ## Response to Q1.
>
> LPCA and ESS used the same total number of samples as the score-model training set, namely 500k. We apologize that this was not stated clearly in the manuscript. The $N=2000$ is the number of samples used for the evaluation.
> We will revise the manuscript to make this comparison protocol explicit.
>
> ---
> ## Response to Q2.
>
> Recent papers report that kNN-based estimators such as LPCA and ESS deteriorate substantially as the ambient dimension increases [5–7]. Levina and Bickel [8] also provide theoretical analysis for a classical kNN estimator, deriving asymptotic bias/variance approximations and noting that high-dimensional settings require very large sample sizes. However, these works do not establish a general theorem that all kNN-based LID estimators must fail in high dimensions.
>
> We will therefore revise the manuscript to replace the phrase “affected by the curse of dimensionality” with the narrower statement that prior kNN-based LID estimators have been observed empirically to deteriorate in higher-dimensional regimes [5–7], with partial theoretical support from classical analyses such as [8].
> To our knowledge, no standard sharp theoretical convergence-rate result is known for the kNN-based LID estimators considered here.
>
> - [5] Tempczyk et al., “Why We Need New Benchmarks for Local Intrinsic Dimension Estimation,” ICLR 2026.
> - [6] Kamkari et al., “A Geometric View of Data Complexity,” NeurIPS 2024.
> - [7] Tempczyk et al., “LIDL: Local Intrinsic Dimension Estimation Using Approximate Likelihood,” ICML 2022.
> - [8] Levina and Bickel, “Maximum Likelihood Estimation of Intrinsic Dimension,” NeurIPS 2004.
>
> ---
> ## Response to Q3.
>
> We did consider whether NB could be accelerated by randomized SVD (rSVD). However, rSVD mainly reduces the linear-algebra cost and does not remove the main bottlenecks of NB. First, NB does not simply require the leading singular vectors; it estimates the normal-space dimension by detecting a drop, or effective numerical rank, in the singular-value spectrum, which is not the setting where standard randomized SVD is most natural. Second, while our main text emphasized the ( O(D^3) ) SVD cost as a clear scalability bottleneck of NB, this is not the only cost: NB must still compute many score vectors around each query point to form the score matrix, and this cost remains substantial even if the SVD step is approximated.
> Finally, in our experiments, NB already showed poor accuracy even with the exact decomposition, so we did not pursue an additional approximation step.
>
> We will revise the manuscript to make this point more balanced: approximate SVD variants may improve the linear-algebra step, but in our view they would not remove the main computational and empirical limitations of NB in our setting.

---

> > ### Author Rebuttal · Reviewer_FRpa · 2026-04-02
> >
> > Thanks to the authors for the detailed and helpful comments, I will raise my score accordingly.

---

### Official Review · Reviewer_aLeL · 2026-03-12

**Soundness:** 3
**Presentation:** 3
**Significance:** 3
**Originality:** 3
**Overall Recommendation:** 4
**Confidence:** 3

**Summary:**

This paper proposes LHSD, a diffusion-based estimator of local intrinsic dimension (LID) based on the spectrum of the local log-density Hessian. The main idea is that tangent directions correspond to small Hessian eigenvalues, while normal directions have large eigenvalues, so LID can be estimated by softly counting near-zero eigenvalues. The method uses a noise-adaptive spectral filter and Stochastic Lanczos Quadrature for scalability.

**Compliance With Llm Reviewing Policy:**

Affirmed.

**Final Justification:**

The authors’ rebuttal was thoughtful and clarified the scope and limitations of the method, but it did not materially resolve my main concerns. In my post-rebuttal update, I noted four specific issues: (Q1) the dependence of LHSD on the Jacobian of the learned score and the heuristic nature of the tangent/normal separation argument under model error; (Q2) the qualitative nature of the discussion of weak spectral-gap failure modes; (Q3) the limited but useful clarification on parameter stability, especially the stronger sensitivity to diffusion time and the heuristic status of the transition-mass diagnostic; and (Q4) the fact that the suggested validations of tangent directions were proposed as future checks rather than evidence included in the paper. The authors did not provide a further response to these points. As a result, my concerns remain unchanged. Overall, I still view the paper as a meaningful and original contribution with strong synthetic evidence, but with unresolved concerns about second-order score fidelity, spectral-gap dependence, and robustness on real data. I therefore maintain my original weak accept.

**Key Questions For Authors:**

1) How sensitive is LHSD to score-model quality, especially the accuracy of second-order derivatives?

2) What are the main failure modes when the spectral gap between tangent and normal directions is small or absent?

3) How stable are the results across the choice of diffusion time and filter parameters on both synthetic and real datasets?

4) How can the authors validate the tangent-space directions implied by the small-eigenvalues used in LID estimation?

**Limitations:**

The paper’s main stated limitation is that LHSD depends on the quality of the trained score model. The authors say that even if training loss converges and generated samples overlap well with the data distribution, this still may not guarantee that the model recovers the true Hessian spectral geometry needed by LHSD, especially on complex real-world data.

*Additional limitations*:

1) *Sensitivity to second-order accuracy*. LHSD needs not just a good score estimate, but a good Jacobian/Hessian of that score field. That is restrictive.

2) *Dependence on spectral separation*. The method is reliable when there is a visible spectral gap between tangent and normal directions; the paper’s own parameter-checking procedure is built around verifying that the filter transition band lies inside that gap. If the gap is small or there is no gap, the estimate may be less trustworthy.

3) *Hyperparameter dependence*. The performance still depends on choosing $(c, p, t)$ appropriately, and there is no principled way of choosing those parameters.

4) *Real-data uncertainty*. The authors show good synthetic results, but they are no results for complex real-world data.

**Strengths And Weaknesses:**

*Soundness*: The method is well motivated, and the Hessian-spectrum view is coherent. The main concern is its reliance on accurate second-order score geometry, which may be hard to guarantee in practice.

*Presentation*: The paper is clear, well structured, and easy to follow from intuition to method to experiments.

*Significance*: The problem is important because of the failure mode of prior methods. The real-data evidence is promising but still limited.

*Originality*: The main idea is to estimate LID by spectrally counting near-zero Hessian eigenvalues. It is novel and different from prior approaches.

*Weaknesses*:

1) The method depends heavily on the second-order accuracy of the learned score model.

2) The approach relies on spectral separation between tangent and normal directions. This assumption can be met in synthetic settings but is not easy to satisfy on real data.

3) The theoretical robustness discussion is good. I feel there is a lack of rigorous development under realistic model.

4) The image experiments are good, but they do not establish the method’s reliability on real data.

5) Some hyperparameter dependence remains, especially through the diffusion time and filter cutoff.

---

> ### Author Rebuttal · Authors · 2026-03-28
>
> We thank the reviewer for the thoughtful and constructive comments. We especially appreciate the emphasis on the practical limitations of second-order score accuracy, spectral separation, and real-data reliability, which helped us clarify the scope and limitations of our method.
>
> ---
>
> ## Q1. How sensitive is LHSD to score-model quality, especially the accuracy of second-order derivatives?
>
> LHSD is sensitive to score-model quality, since it relies on the Jacobian of the learned score. For the true score, $-\nabla_x s(x)$ coincides with the Hessian of $-\log p(x)$ and is therefore symmetric. For a learned score field, however, $-\nabla_x s_\theta(x)$ need not be symmetric; this non-symmetry reflects a non-conservative component of the learned vector field.
>
> Such non-conservative effects may affect the interpretation of individual eigenvalues. However, [3] suggests that they tend to appear mainly as within-manifold mixing, while the normal-direction structure is comparatively less affected. This suggests that the coarse tangent/normal spectral separation used by LHSD may be more stable than the exact eigenvalues themselves, although this is only a heuristic and may fail when the spectral gap is weak.
>
> - [3] Wenliang et al., "Score-based generative models learn manifold-like structures with constrained mixing."
>
> ---
> ## Q2. What are the main failure modes when the spectral gap between tangent and normal directions is small or absent?
>
> Several situations can be considered.
> 1) If the density along the manifold is highly non-uniform, tangent eigenvalues may no longer cluster tightly near zero.
> 2) If the local geometry is complicated, for example highly curved or close to nearby structures, the distinction between tangent and normal directions can become ambiguous.
> 3) Insufficient score learning, especially in second-order derivatives, can artificially blur the spectrum by shifting tangent modes upward and dispersing normal modes.
>
> We expect such effects to be more common on real datasets, where the spectral-gap assumption may hold only approximately.
>
> ---
>
>
> ## Q3. How stable are the results across the choice of diffusion time and filter parameters on both synthetic and real datasets?
>
> In our current experiments, LHSD was relatively insensitive to the filter parameters $c$ and $p$ within a reasonable range, but more sensitive to the diffusion time $t$.
>
> For small noise, tangent-related eigenvalues remain near $0$, while normal-related eigenvalues appear around $1/\sigma(t)^2$. Therefore, as long as the cutoff $\kappa(t)=c/\sigma(t)^2$ lies inside the spectral gap, the estimate is fairly insensitive to moderate changes in $c$ and $p$. In our synthetic experiments, values such as $c=0.1$ or $0.2$ worked consistently across dimensions.
>
> By contrast, $t$ is more critical: if $t$ is too large, smoothing weakens the spectral gap and the estimate becomes less stable. On real data, where the gap may be less clean, this issue is more pronounced. We therefore use the transition-mass diagnostic as a practical heuristic for identifying a reliable range of $t$, rather than as a fully principled parameter-selection rule.
>
> ---
>
> ## Q4. How can the authors validate the tangent-space directions implied by the small-eigenvalues used in LID estimation?
>
> Several post-hoc validations are possible. Using the same sign convention as in LHSD, let $H(x,t) = -\nabla^2 \log p_t(x)$ (or its learned approximation $-\nabla_x s_\theta(x,t)$).
>
> 1. **Score alignment.** For small noise, the score $s_\theta(x,t)$ is expected to be approximately orthogonal to the manifold and thus aligned mainly with normal directions [4]. Therefore, if a direction $v$ satisfies small $|s_\theta(x,t)^\top v|$, it is more likely to be tangent, whereas a larger value suggests a stronger normal component.
>
> - [4] Stanczuk et al. "Diffusion models encode the intrinsic dimension of data manifolds." ICML 2024.
>
> 2. **Perturbation stability.** The quadratic form $v^\top H(x,t)v$ measures curvature along $v$. One can test whether it remains small when perturbing the point $x$ slightly and when varying the noise level $t$ locally. Tangent directions should stay relatively flat under both types of perturbation, while normal directions should show larger curvature and more clearly follow the $1/\sigma(t)^2$ scaling.
>
> 3. **Known infinitesimal transformations.** For image data, one can construct approximate tangent vectors induced by small translations, rotations, or brightness changes, and check whether they project strongly onto the near-zero eigenspace of $H(x,t)$. A large projection would support the interpretation that this eigenspace captures tangent directions.

---

> > ### Author Rebuttal · Reviewer_aLeL · 2026-04-03
> >
> > **Q1.** The authors confirm that LHSD depends on the Jacobian of the learned score and acknowledge non-conservative effects. Their argument that coarse tangent/normal separation may remain stable is interesting, but explicitly heuristic. This clarifies the limitation but does not resolve it.
> >
> > **Q2.** The authors provide a clear list of failure modes when the spectral gap is weak: non-uniform manifold density, complex local geometry, nearby structures, and imperfect second-order score learning. This is helpful, but remains qualitative and does not reduce the concern.
> >
> > **Q3.** The authors clarify that LHSD is fairly stable to $(c, p)$ but more sensitive to diffusion time $t$, and they appropriately describe the transition-mass diagnostic as a heuristic rather than a principled rule. This improves transparency, but only modestly.
> >
> > **Q4.** The proposed validations of tangent directions are sensible and specific, but they are future checks rather than evidence included in the paper. So, this does not strengthen the current submission.
> >
> > *Conclusion: The rebuttal is good and clarifies scope well, but it does not change my evaluation.*

---

### Official Review · Reviewer_Y8s8 · 2026-03-13

**Soundness:** 2
**Presentation:** 2
**Significance:** 2
**Originality:** 2
**Overall Recommendation:** 5
**Confidence:** 4

**Summary:**

This work proposes a novel method for estimating the intrinsic dimension of a manifold locally. The proposed method, named Local Hessian Spectral Dimension (LHSD), conceptually relies on a spectrally filtered version of the log-density Hessian,. The authors show that the resulting estimator is robust to noise and errors emerging from the dimensionality of the ambient space. The proposed estimator is also scalable as its complexity is linear to the dimension of the ambient space. Furthermore, the authors provide a visual process under which hyper-parameter selection can performed effectively. At last, the authors provide extended convincing experimental results after considering real and synthetic datasets.

**Compliance With Llm Reviewing Policy:**

Affirmed.

**Final Justification:**

After considering the rebuttal and subsequent discussion, my assessment has improved. The authors addressed most of my initial concerns satisfactorily. While the post-rebuttal responses were less helpful, partly due to time constraints or points of disagreement, these issues do not affect the core contributions of the paper. There is general agreement among the reviewers that the work is mathematically sound, clearly presented, and well motivated, particularly in relation to known limitations of prior approaches. Although the problem itself is not central, it is currently an active area of research. Some weaknesses remain, as noted in my review and in the comments of reviewer aLeL. However, I believe the proposed contributions can be valuable to the community and relevant practitioners in their current form, and even more so after incorporating improvements addressing points raised by all reviewers in the final version. Based on this, I increase my score to 5.

**Key Questions For Authors:**

Question 1: Why may increasing m degrade the performance of your method? In the experiments, this appears to happen in some cases even for linear manifolds. Also, what happens when m exceeds 5?

Question 2: The metric used in the experiment 1 does not reveal whether the error distribution is uniform across the three manifolds or skewed more toward one of them. Have you observed any such imbalance in practice?

Question 3: If Π_nor denotes an orthogonal projection matrix, then its eigenvalues should be binary. I do not understand how the decomposition property stated in Property 1 follows from this. Could you clarify this point?

Question 4: A limitation of your method appears to be its reliance on the quality of the trained score model, according to you. In the experiments, you mention using 500k samples for training, which I assume refers to training the score function. This is a significant number of samples that may not be available in many practical applications. Could you comment on the sensitivity of your method to the amount of training data?

Question 5: Methods such as Hessian Eigenmaps proposed by Donoho and Grimes can also be used in theory to estimate intrinsic dimensionality locally. What is the conceptual advantage of your method compared to such approaches?

**Limitations:**

Yes

**Strengths And Weaknesses:**

The submission seems to be sound, as well as the supported claims both from theoretical and experimental aspect. Minor exceptions have  raised some questions below. The used methods are appropriate and the considered experiments are well-designed. Finally, the authors seem to be careful and honest about evaluating both the strengths and weaknesses of their work.

Overall, the paper is well written and well structured. There are some parts that could be improved. For example, Diffusion-based LID Estimation seems to be the only alternative line on research on the problem under consideration. However, I'm not sure that this is the case. A more detailed or definite description of related works would be useful, as based on this work score-based diffusion models seem to be the only promising direction for this problem. That said, the author position their method properly in the context of prior/concurrent literature on this direction and clearly discuss how it differs. Other than that, the rest of the presented material is quite detailed, and the overall narrative easy to follow.

The paper addresses a relevant problem and considers a motivating application that further highlights its importance. The work is primarily practical in nature, as it proposes an effective method for tackling the problem. Moreover, the fact that the method can be easily tuned through visual inspection makes it particularly appealing for practitioners. As such, it could have an impact on the community, although its scope is somewhat narrow. Nevertheless, in my opinion, it still represents an appropriate and meaningful contribution.

In terms of originality,  the work does provide new insights, highlights important properties of existing methods, and introduces new methods and perspectives that advance the field in some dimensions. The contributions clearly distinguished from closely related literature, and is the novelty justified sufficiently well. Potential minor objections on the latter one have been raised in the questions.

---

> ### Author Rebuttal · Authors · 2026-03-28
>
> We thank the reviewer for the careful reading and comments, which improved the paper’s clarity.
>
> ---
>
> ## Response to Q1.
>
> As discussed in Appendix H.1, increasing $m$ beyond 5 yields little further gain.
> One plausible reason $m=10$ or $20$ can underperform $m=5$ is **finite-precision loss of orthogonality** in the Lanczos basis. In exact arithmetic, the Lanczos basis vectors $q_1,\dots,q_m$ should remain orthonormal, i.e., $q_i^\top q_j=\delta_{ij}$. In finite precision, however, this orthogonality can gradually deteriorate, so later vectors may partially re-enter previously discovered directions.
> This can produce duplicated or spurious Ritz values [2] and make $e_1^\top f(T_m)e_1$ less stable, degrading downstream MAE.
> Possible mitigations include higher precision and full reorthogonalization [1].
>
> - [1] Chen et al., “Analysis of stochastic Lanczos quadrature for spectrum approximation,” ICML, 2021.
> - [2] Meurant and Strakoš, “The Lanczos and conjugate gradient algorithms in finite precision arithmetic,” Acta Numerica, 2006.
>
> Second, for the linear manifolds in Table 1, where tangent/normal separation is relatively clean, even small $m$ may already capture the main structure needed for LID estimation; in that case, larger $m$ can become more sensitive to score, HVP, and numerical errors, slightly worsening MAE.
>
> ---
>
> ## Response to Q2.
>
> We agree that the aggregate metric in Experiment 1 does not reveal whether errors are balanced across the three manifolds.
> From the histograms in Figs. 4 and 15, we
> did not observe a systematic imbalance in which one manifold dominated the error. The main trends were: 1) increased variance with ambient/manifold dimension, and 2) a slight downward bias on nonlinear manifolds (e.g., Fig. 15g vs. Fig. 15f).
>
> ---
> ## Response to Q3.
> The binary eigenvalues $(0,1)$ belong to the projector $\Pi_{\mathrm{nor}}$ itself, not to the Hessian $H(x)$. Here $p_t$ is the density after Gaussian smoothing by the diffusion process. Locally near the manifold, its log-density can be written as
> $
> \log p_t(x) \approx \log p_{\mathcal M}(x_\parallel) - \frac{\|x_\perp\|^2}{2\sigma(t)^2} + O(1),
> $
> where $x_\parallel$ and $x_\perp$ denote the tangent and normal components.
>
> The term $-\|x_\perp\|^2/(2\sigma(t)^2)$ is the log-density of a Gaussian with variance $\sigma(t)^2$ in the normal directions, up to an additive constant. Taking the Hessian with respect to the normal coordinates gives
> $
> -\nabla_\perp^2 \left(- \frac{\|x_\perp\|^2}{2\sigma(t)^2}\right)
> = \frac{1}{\sigma(t)^2} I_\perp,
> $
> which gives the scale $1/\sigma(t)^2$.
>
> Therefore,
> $
> H(x) = -\nabla^2 \log p_t(x)
> = \frac{1}{\sigma(t)^2}\Pi_{\mathrm{nor}}(x) + O(1).
> $
> Since $\Pi_{\mathrm{nor}}$ acts as $0$ on the tangent subspace and $1$ on the normal subspace, the leading-order eigenvalues of $H$ are near $0$ and $1/\sigma(t)^2$. Thus, after absorbing the bounded remainder into the $O(1)$ term, this yields
> - $\lambda_i = O(1)$ for tangent directions,
> - $\lambda_j = \frac{1}{\sigma(t)^2} + O(1)$ for normal directions,
>
> which is the statement of Property 1.
>
> We also noticed a typo in the manuscript:
> $
> \lambda_j + \frac{1}{\sigma(t)^2} + O(1)
> $
> should read
> $
> \lambda_j = \frac{1}{\sigma(t)^2} + O(1).
> $
> We apologize for this mistake, which may have contributed to the reviewer’s confusion.
>
> ---
>
> ## Response to Q4.
>
> Yes, we used 500k samples to train the score model. This was not a requirement of LHSD itself, but an experimental choice to reduce score-model error and isolate the behavior of the LID estimators. We agree that LHSD depends on score quality: reducing training data can degrade the score, blur tangent/normal spectral separation, and lower estimation accuracy.
> While we did not perform a dedicated training-set-size ablation, Sec. 4 suggests that LHSD is relatively robust to score approximation error compared with existing baselines.
>
> ---
>
> ## Response to Q5.
>
> Hessian Eigenmaps (HLLE) is conceptually different from LHSD: it is not a pointwise local intrinsic-dimension estimator, but a method that assumes the manifold dimension $d$ and seeks a $d$-dimensional parametrization through the low-curvature function space associated with the isometric coordinates. Still, it is relevant to LHSD because both rely on second-order curvature and local tangent flatness. We will clarify this briefly in Sec. 2.3.
>
> ---
>
> ## Response to the concern: Is Diffusion-based LID Estimation the only alternative line?
> Diffusion-based methods are not the only conceivable route to LID estimation: in principle, any generative model that provides access to a high-dimensional density or its score could serve as a basis. For example, LIDL was originally instantiated with normalizing flows. Our point was only that diffusion-based methods currently constitute the main established scalable parametric line in the LID literature; flow matching is another natural candidate, but, to the best of our knowledge, there is not yet an established LID estimator built on top of it.

---

> > ### Author Rebuttal · Reviewer_Y8s8 · 2026-04-02
> >
> > The authors have satisfactorily addressed almost all my questions. Regarding Q3, it was indeed the typo that confused me. Regarding Q4, I remain skeptical. Although there are theoretical claims on robustness for your method (section 4), you still chose to proceed with half a million training samples. An ablation study in this direction would definitely strengthen your work. Finally, my understanding is that Hessian Eigenmaps can also be used for determining the global intrinsic dimension of a manifold. If the samples are drawn from a neighborhood, then I understand that it could be used for determining LID. Is there anything I'm missing here? If not, what would be the disadvantage of this approach over yours?

---

> > > ### Author Response · Authors · 2026-04-03
> > >
> > > We thank the reviewer for the follow-up comment and for giving us the opportunity to make this point clearer.
> > >
> > > ---
> > >
> > > **Regarding Q4: Dependence on training set size**
> > >
> > > We agree that a training-set-size ablation would strengthen the paper, and we will add it in the revision.
> > >
> > > As preliminary evidence during the rebuttal period, we reduced the training set from 500k to 50k samples on the $1024$-dimensional $\{10+80+200\}$ manifold mixture in both Linear and Nonlinear settings. Because of time constraints, the reduced-data runs were trained for only 160 epochs, whereas the original experiments were trained for 400 epochs. This comparison is therefore preliminary and reflects both reduced sample size and a shorter training budget. We use training loss as a practical proxy for score quality.
> > >
> > > The results are:
> > >
> > > | Setting | Train samples | Epochs | Training loss | MAE |
> > > |---|---:|---:|---:|---:|
> > > | Linear | 500k | 400 | 57.51 | 3.47 |
> > > | Linear | 50k | 160 | 84.33 | 10.34 |
> > > | Nonlinear | 500k | 400 | 135.63 | 6.90 |
> > > | Nonlinear | 50k | 160 | 151.18 | 6.93 |
> > >
> > > In this preliminary test, a large worsening in training loss, as in the linear setting (about 46.6%), was accompanied by a clear increase in MAE. In contrast, in the nonlinear setting, where the training loss worsened by about 11.5%, the MAE changed only marginally. We will add a more systematic and controlled ablation in the revised paper.
> > >
> > > ---
> > >
> > > **Regarding Q5: Hessian Eigenmaps (HLLE)**
> > >
> > > HLLE strictly requires the true or assumed manifold dimension, $d$, as an a priori hyperparameter. For each sample, it performs a neighborhood SVD in the $D$-dimensional ambient space and extracts the top $d$ eigenvectors to form local coordinates on the $d$-dimensional manifold. Using HLLE itself to estimate $d$ would therefore be methodologically circular.
> > >
> > > We further note that this neighborhood SVD step in HLLE is mathematically equivalent to Local PCA (LPCA), which we already evaluated as a baseline for Local Intrinsic Dimension (LID) estimation in our experiments.

---

### Decision · Program_Chairs · 2026-04-30

**Decision:**

Accept (regular)

**Comment:**

Reviewers all agree the paper is well written, well structured, and the argumentation is coherent and sound. The proposed method is sensible and shows potentially very good practical utility for an important and challenging problem. The authors also did a very good job in the rebuttal phase to allay the main concerns, suggesting the revised paper will improve meaningfully on the initial submission.